# Mgl2+ cDC2s coordinate fungal allergic airway type 2, but not type 17, inflammation in mice

Peter C. Cook [1,2] ✉, Sheila L. Brown[2], Emma L. Houlder [2], Julio Furlong-Silva[1], Daniel P. Conn [1], Stefano A. P. Colombo[2], Syed Baker [2], Freya R. Svedberg[2], Gareth Howell[2], Margherita Bertuzzi [3], Louis Boon [4], Joanne E. Konkel [2], Christopher R. Thornton[5], Judith E. Allen [2] & Andrew S. MacDonald [2,6] ✉

Fungal spores are abundant in the environment and a major cause of asthma. Originally characterised as a type 2 inflammatory disease, allergic airway inflammation that underpins asthma can also involve type 17 inflammation, which can exacerbate disease causing failure of treatments tailored to inhibit type 2 factors. However, the mechanisms that determine the host response to fungi, which can trigger both type 2 and type 17 inflammation in allergic airway disease, remain unclear. Here we find that CD11c+ DCs and CD4+ T cells are essential for development of both type 2 and type 17 airway inflammation in mice repeatedly exposed to inhaled spores. Single cell RNA-sequencing with further multi-parameter cytometry shows that allergic inflammation dramatically alters the proportion of numerous DC clusters in the lung, but that only two of these (Mgl2+ cDC2s and CCR7+ DCs) migrate to the dLNs. Targeted removal of several DC subsets shows that Mgl2+ cDC2 depletion reduces type 2, but not type 17, fungal allergic airway inflammation. These data highlight distinct DC subsets as potential therapeutic targets for the treatment of pulmonary fungal disease.

Asthma causes a huge burden on global public health, with ~300 million sufferers worldwide, and incidence rapidly rising[1]. Fungal spores are abundant in our environment and are potent, yet often overlooked, triggers of the allergic inflammation that underpins asthma[2]. Fungal sensitisation in the airways, especially to inhaled *Aspergillus fumigatus* (*Af*), is known to increase the risk of developing significant asthmatic disease, such as allergic bronchopulmonary aspergillosis (APBA) and severe asthma with fungal sensitisation (SAFS)[3]. These conditions are estimated to impact at least 10 million people worldwide[4]. Despite this, the mechanisms that lie behind fungal initiation of these diseases are poorly understood, which reduces our ability to develop improved diagnostics and therapeutics.

Immune cells are central to the inflammation underlying asthma, which was originally characterised as a disease dominated by type 2 cytokines (e.g. IL-4, IL-5 and IL-13) leading to granulocyte activation (especially eosinophils and mast cells), airway hyper-responsiveness (AHR), mucus overproduction and fibrosis[5]. However, recent evidence has shown that some asthma sufferers display low type 2 responses, instead presenting with prominent neutrophilia and type 17 cytokines[6,7]. These distinct inflammatory pathways (type 2 high/eosinophilic vs type 2 low/neutrophilic) can be used to classify 'endotypes' of patients, each underpinned by different immune characteristics. This may be vital when considering therapeutic strategies against specific aspects of the immune response in asthma, especially given

[1]Medical Research Council Centre for Medical Mycology at the University of Exeter, Department of Biosciences, Faculty of Health and Life Sciences, Geoffrey Pope Building, Stocker Road, Exeter, United Kingdom. [2]Lydia Becker Institute of Immunology and Inflammation, University of Manchester, Manchester, United Kingdom. [3]Manchester Fungal Infection Group, University of Manchester, Manchester, United Kingdom. [4]JJP Biologics, Warsaw, Poland. [5]Department of Biosciences, Faculty of Health and Life Sciences, University of Exeter, Exeter, United Kingdom. [6]Institute of Immunology and Infection Research, University of Edinburgh, Edinburgh, United Kingdom. ✉e-mail: p.c.cook@exeter.ac.uk; andrew.macdonald@ed.ac.uk

the disappointing efficacy thus far of therapeutics that target type 2/eosinophilic inflammation in some asthmatics[5,8]. In the context of fungal infection more generally, type 17 inflammation is a crucial aspect of protective immunity[9], but constant exposure to fungi (including *Af*) has long been known to also induce significant levels of type 2 inflammation[10–12]. However, the underlying immune events that play a major role in coordinating type 2/eosinophilic vs type 17/neutrophilic fungal inflammation are currently poorly understood.

Several major cell types have been proposed as the dominant sources of type 2 and type 17 cytokines that promote downstream disease in asthma. It is apparent that type 2 and type 17 cytokines are not just secreted by Th2 or Th17 CD4⁺ T cells, but can also be produced by innate immune cell populations such as innate lymphoid cells (ILCs) and γδ T cells[5,13,14]. Although the majority of studies so far have focused more on non-fungal allergens such as house dust mite (HDM) extract, fungi (e.g. *Alternaria*) and fungal cell wall components (e.g. chitin) have been found to initiate allergy in the lung by activating IL-13 secreting ILC2s via epithelial damage and release of alarmin cytokines such as TSLP and IL-33[15,16]. In the context of fungal allergic inflammation, few studies have assessed the relative contribution of different cell types in promoting responses that underpin disease. However, inflammation is accompanied by CD4⁺ T cell secretion of type 2 and type 17 cytokines in human ABPA patients[17], suggesting that CD4⁺ T cells may be integral to the development of allergic airway disease.

Although it's well established that CD4⁺ T cell mediated allergic inflammation is dependent on antigen presenting cells, especially dendritic cells (DCs)[18,19] the role of DCs in co-ordinating type 2 vs type 17 endotypes of allergic inflammation is poorly understood. There are several subsets of lung DCs, categorised based on their ontogeny, development and functional capabilities[20]. Currently, little is known about the relative contribution of distinct DC subsets in induction of fungal allergic inflammation[21]. In the context of invasive fungal disease (aspergillosis), plasmacytoid DCs (pDCs) and monocyte-derived DCs (moDCs) have been attributed to coordinating protective immunity[22,23]. Influx of moDCs has also been implicated in regulating type 2 and type 17 inflammation in response to persistent fungal infection[24]. Different subsets of conventional DCs (cDCs) have been linked to distinct inflammatory outcomes; cDC1s (associated with high level IL-12 production, cross-presentation and priming of type 1 inflammation against tumour antigens, bacteria, viral and protozoal pathogens[25–28]) have been proposed to dampen allergic inflammation and govern anti-fungal immunity (especially type 17)[29,30], whereas cDC2s have been implicated in promotion of protective anti-fungal type 2 and type 17 inflammation[31,32]. Furthermore, cDC2s have previously been shown to induce allergic airway inflammation in response to HDM[19]. To our knowledge, despite the crucial role that DCs have in promoting immunity in diverse disease settings, the role of these different subsets in coordinating fungal allergic airway inflammation has not yet been fully addressed.

In this work, we use a mouse model of direct, intranasal (i.n.), sensitisation and challenge with repeated low doses of live *Af* conidia to induce allergic airway inflammation with both type 2 and type 17 cytokine and cellular features, including airway eosinophilia and neutrophilia. Through use of cytokine reporter mice, we show this is accompanied by an increase of type 2 (IL-13⁺) and type 17 (IL-17A⁺) cytokine expressing cells in the airways, lung tissue and lung draining lymph nodes (LNs), with CD4⁺ T cells, not innate cells (e.g. γδ T cells and ILCs), the dominant sources of these cytokines. Further, CD4⁺ T cells and CD11c⁺ antigen presenting cells (APCs) are essential for development of fungal type 2 and type 17 allergic airway inflammation. Combining use of single cell RNA-sequencing (scRNAseq), flow cytometry and mass cytometry, we show that Mgl2 expressing cDC2s expand in the lungs of *Af*-exposed mice, and are the major DC population in the lung draining LNs, during fungal allergic airway inflammation. Importantly, depletion of Mgl2⁺ cells impairs fungal type 2 allergic airway inflammation, while failing to significantly affect type 17 readouts. In contrast, cDC1 deficiency or pDC depletion has minimal impact on either type 2 or type 17 responses. Overall, use of complementary single cell technologies reveals that a specific DC subset, namely Mgl2⁺ cDC2s, are critical in mediating in vivo development of type 2, but not type 17, features of fungal allergic airway inflammation.

## Results

### Repeat fungal spore exposure induces a mixed type 2/type 17 pulmonary inflammatory response

The mechanisms that determine the character of the host response to fungi, which can trigger both type 2 and type 17 inflammation in allergic airway disease, remain unclear[2,21]. To directly address this, we exposed mice i.n. to live *Af* spores over a 19 day period, sampling the airways (via BAL isolation) and lung tissue after 3, 6 or 9 repeat doses (Fig. 1A). In contrast to what has been observed with mouse models of invasive aspergillosis, where there is rapid monocyte/neutrophil influx after a single exposure of immunocompromised mice to a high dose of *Af* spores[22], 3 low doses of *Af* spores to immunocompetent C57BL/6 mice induced only minor neutrophilia (d5 after the initial exposure) (Fig. 1B and Supplementary Fig. 1A). However, after 6 or 9 spore doses (d12 and d19, respectively), both eosinophils and neutrophils, hallmarks of type 2 and type 17 inflammation, were prominent in the BAL fluid and lung tissue (Fig. 1B and Supplementary Fig. 1A, B). These responses were accompanied by low fungal colonisation throughout (CFUs were ~0.02% of total administered *Af* conidia (Supplementary Fig. 1C). In the BAL fluid we detected factors associated with type 1/type 17 (CXCL1, IL-6, IL-12p40) and type 2 (amphiregulin (Areg), CCL17, CCL22, RELMα and Ym1) inflammation after 6 or 9 spore doses (Supplementary Fig. 1D–F). We did not see significantly increased IFNγ in the BAL fluid of *Af*-exposed mice (Supplementary Fig. 1D), or reliable levels of some type 17 and type 2 cytokines, including IL-33 and TSLP (which have been proposed to be important factors for boosting ILC and CD4⁺ T cell mediated allergic inflammation[15,33]) (Supplementary Fig. 1D–F). However, we detected increased mRNA expression for several of these mediators in lung tissue (Supplementary Fig. 1G). These cytokine changes were accompanied by increased BAL lymphoid cells, including γδ, CD4⁺ and CD8⁺ T cells, as well as B cells (Fig. 1C and Supplementary Fig. 2A). Furthermore, *Af*-specific serum IgG was significantly elevated in mice exposed to *Af*, but only following 9 spore doses (Supplementary Fig. 2B).

To simultaneously identity the main type 2 and type 17 cytokine expressing cells responding in the lung to *Af* exposure, we generated dual IL-13/IL-17 reporter mice by crossing *Il13*^eGFP with *Il17*^CreROSA^eYFP[34,35] mice. Surprisingly, in contrast to other models of fungal airway inflammation such as *Alternaria*[36], there were very few IL-13^eGFP+ cells detected at the early stages of *Af* exposure (d5) in either the BAL fluid, lung tissue or lung dLNs (Fig. 1D and Supplementary Fig. 3A). Neither was a significant increase in the number of IL-17^eYFP+ cells detected at this timepoint (Fig. 1D and Supplementary Fig. 3A). However, by d12 to d19, a striking increase in IL-13^eGFP+ and IL-17^eYFP+ cells was evident in the BAL fluid, lung tissue and dLNs of *Af*-exposed mice.

Using histology of lung sections to assess the impact of these type 2 and type 17 responses on lung pathology after spore exposure, we found features associated with allergic inflammation, such as increased epithelial thickness (following 6 or 9 spore doses) and increased goblet cell numbers (after 9 spore doses) but not collagen deposition (Supplementary Fig. 4A–C). Together, these data demonstrate that repeat exposure to *Af* spores simultaneously induces hallmark features of both type 2 and type 17 allergic inflammation in the airways, lung tissue and lung dLNs.

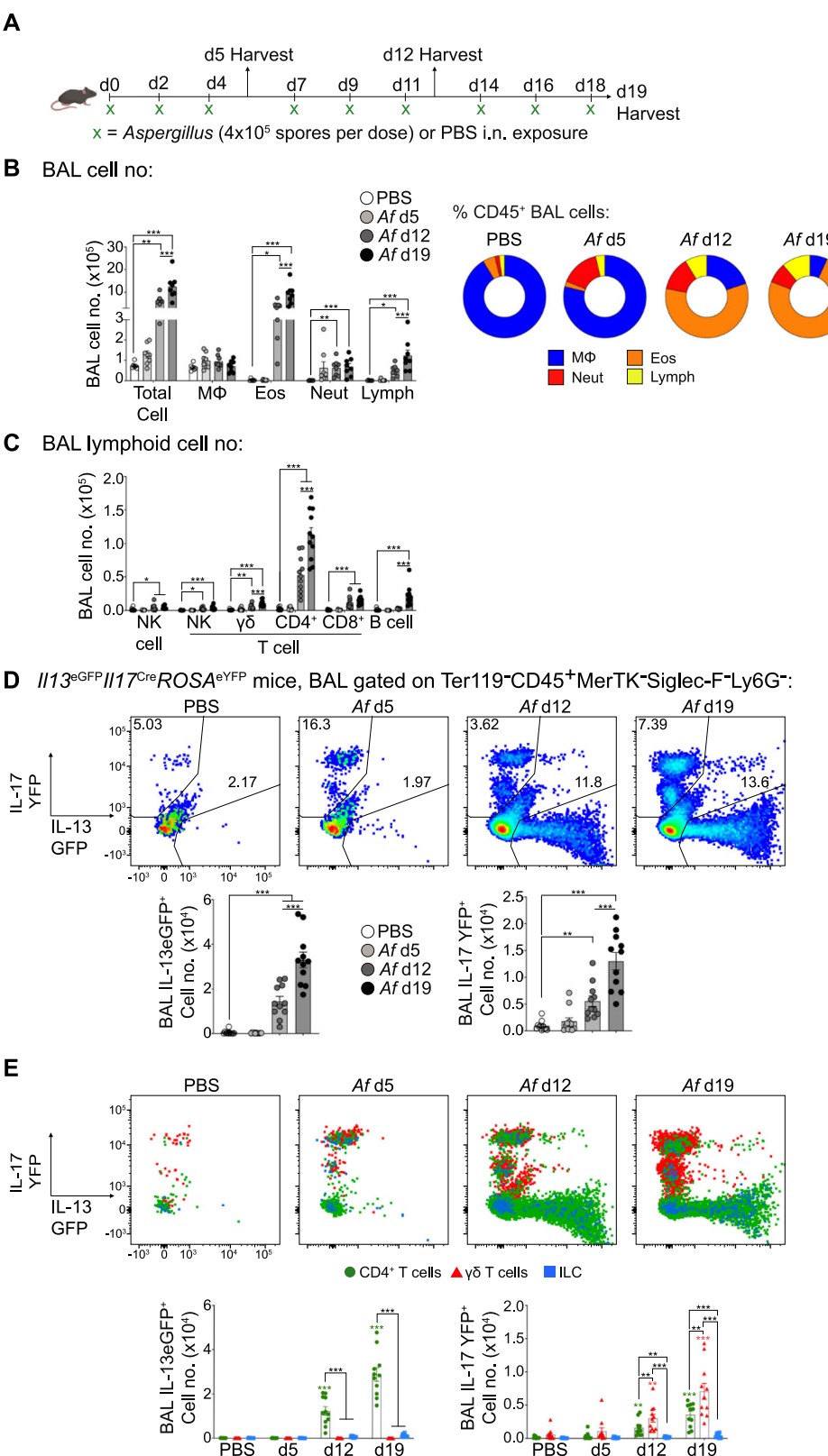

**CD4⁺ T cells and γδ T cells are the primary sources of type 2 and type 17 cytokines in the pulmonary fungal allergic response**

A variety of cell types, including CD4⁺ T cells, γδ T cells, ILCs, and even granulocytes, have been proposed to be important sources of type 2 or type 17 cytokines during pulmonary inflammation against a variety of allergens, including fungi[37,38]. However, we did not observe notable IL-13eGFP or IL-17eYFP expression by eosinophils or neutrophils in *Af*-exposed mice (Supplementary Fig. 3B). Analysis of IL-13eGFP⁺ cells in the BAL fluid, lung tissue and lung dLNs revealed that the overwhelming majority were CD4⁺ T cells, with surprisingly few ILCs, and minimal γδ T cells (Fig. 1E and Supplementary Fig. 3C). Assessment of IL-17eYFP⁺ cells following *Af* spore exposure identified a fairly equal split of γδ T cells and CD4⁺ T cells, and negligible ILC positivity, in the BAL fluid, lung tissue or lung dLNs (Fig. 1E and Supplementary Fig. 3C). Further,

**Fig. 1 | Fungal spores induce hallmarks of type 2 and type 17 allergic airway inflammation. A** Mice were repeatedly exposed to PBS or *A. fumigatus* spores (*Af*, strain: CEA10, $4 \times 10^5$ spores per dose) or PBS controls via intranasal transfer on the indicated time points. Tissues were harvested the following day after the third, sixth or ninth dose of spores (d5, d12 and d19, respectively). Created in BioRender. Cook, P. (2024) https://BioRender.com/z24b189. **B** Graphs display the numbers and percentage of macrophages (MΦ), eosinophils (Eos), neutrophils (Neut) and lymphocytes (Lymph) isolated from the BAL fluid of mice following repeat doses of *Af* spores. **C** Graph displays the numbers of NK cells, NK T cells, γδ T cells, CD4⁺ T cells, CD8⁺ T cells and B cells isolated from the BAL fluid of mice following repeat doses of *Af* spores. **D** Flow cytometry plots and graphs identify the proportion and number of *Il13* and *Il17* expressing cells isolated from the BAL fluid of *Il13*ᵉᴳᶠᴾ*Il17*ᶜʳᵉ*ROSA*ᵉʸᶠᴾ mice exposed to varying doses of *Af* spores. **E** Flow cytometry overlay plots of *Il13* and *Il17* expression of CD4⁺ T cells (green), γδ T cells (red), ILCs (blue). Graphs show the number of *Il13* and *Il17* cells for each population that were detected in the BAL fluid of *Il13*ᵉᴳᶠᴾ*Il17*ᶜʳᵉ*ROSA*ᵉʸᶠᴾ mice exposed to varying doses of *Af* spores. **B** data are from 2 independent experiments (*n* = 30 biologically independent animals). **C, D & E** data are from 3 independent experiments (*n* = 44 biologically independent animals). Data were fit to a linear mixed effect model, with experimental day as a random effect variable, and groups compared with a two-sided Tukey's multiple comparison test. *$P < 0.05$, ** $P < 0.01$, *** $P < 0.001$. Data are presented as mean values ± SEM. **B** *P* Values, Total Cell; PBS vs *Af* d12 = 0.0011, PBS vs *Af* d19 = $2.19 \times 10^{-13}$, *Af* d12 vs *Af* d19 = $6.34 \times 10^{-5}$. Eos; PBS vs *Af* d12 = 0.00114, PBS vs *Af* d19 = $2.86 \times 10^{-11}$, *Af* d12 vs *Af* d19 = $3.33 \times 10^{-5}$. Neut; PBS vs *Af* d12 = 0.00268, PBS vs *Af* d19 = 0.0005. Lymph: PBS vs *Af* d12 = 0.03553, PBS vs *Af* d19 = $4.08 \times 10^{-9}$, *Af* d12 vs *Af* d19 = 0.000342. **C** *P* Values, NK; PBS vs *Af* d12 = 0.02673, PBS vs *Af* d19 = 0.01309. NK T cell; PBS vs *Af* d12 = 0.0278, PBS vs *Af* d19 = $9.02 \times 10^{-6}$. γδ T cell; PBS vs *Af* d12 = 0.001323, PBS vs *Af* d19 = $1.36 \times 10^{-13}$, *Af* d12 vs *Af* d19 = 0.000151. CD4⁺ T cell; PBS vs *Af* d12 = $4.53 \times 10^{-8}$, PBS vs *Af* d19 = $< 2 \times 10^{-16}$, *Af* d12 vs *Af* d19 = $6.82 \times 10^{-11}$. CD8⁺ T cell; PBS vs *Af* d12 = $4.864 \times 10^{-6}$, PBS vs *Af* d19 = $5.6 \times 10^{-10}$. B cell; PBS vs *Af* d19 = $2.15 \times 10^{-10}$, *Af* d12 vs *Af* d19 = $5.2 \times 10^{-9}$. **D** *P* Values, IL-13eGFP⁺: PBS vs *Af* d12 = $1.28 \times 10^7$, PBS vs *Af* d19 = $< 2 \times 10^{-16}$, *Af* d12 vs *Af* d19 = $1.93 \times 10^{-12}$. IL-17 YFP⁺: PBS vs *Af* d12 = 0.0024, PBS vs *Af* d19 = $< 2 \times 10^{-16}$, *Af* d12 vs *Af* d19 = $2.11 \times 10^{-7}$. **E** *P* Values, IL-13eGFP⁺: *Af* d12 CD4⁺ T cells vs γδ T cells = $< 2 \times 10^{-16}$, *Af* d12 CD4 + T cells vs ILCs = $2 \times 10^{-16}$, *Af* d19 CD4⁺ T cells vs γδ T cells = $2 \times 10^{-16}$, *Af* d19 CD4⁺ T cells vs ILCs = $2 \times 10^{-16}$. IL-17 YFP⁺: *Af* d12 CD4⁺ T cells vs γδ T cells = 0.0036, *Af* d12 CD4⁺ T cells vs ILCs = 0.0036, *Af* d12 γδ T cells vs ILCs = $3.02 \times 10^{-9}$, Source data are provided as a Source Data File. Schematics in figures were created in https://BioRender.com.

strikingly few cells were double positive for IL-13ᵉᴳᶠᴾ and IL-17ᵉʸᶠᴾ, suggesting that type 2 and type 17 cytokine secreting cells were predominantly separate populations (Fig. 1E). Supporting these reporter data, ex vivo cytokine staining confirmed that CD4⁺ T cells increased expression of IL-4, IL-5, IL-13 and IL-17 (Supplementary Fig. 3D), while γδ T cells displayed elevated IL-17, upon repeat *Af* spore exposure (Supplementary Fig. 3E).

Overall, these data show that CD4⁺ and γδ T cells, not ILCs or granulocytes, are the major sources of type 2 and 17 cytokines during the allergic airway response against *Af* spores.

### CD4⁺ T cells are essential for type 2 and 17 fungal allergic pulmonary response

IL-13 and IL-17 secretion from activated CD4⁺ or γδ T cells has previously been implicated in promoting allergic and/or anti-fungal inflammation[6,39,40]. As we had identified that CD4⁺ T cells were the major population expressing IL-13 or IL-17 in response to *Af* (Fig. 1E and Supplementary Fig. 3C), we used anti-CD4 depleting antibody to target CD4⁺ T cells in *Il13*ᵉᴳᶠᴾ*Il17*ᶜʳᵉ*ROSA*ᵉʸᶠᴾ mice. By d12, anti-CD4 mAb treated mice showed a dramatic reduction of eosinophilia and neutrophilia in the BAL fluid and lung tissue (Fig. 2A and Supplementary Fig. 5A). Furthermore, anti-CD4 mAb treatment reduced lymphoid cell populations in the BAL fluid (including γδ T cells) and successfully depleted CD4⁺ T cells in the lung and dLNs (Fig. 2A and Supplementary Fig. 4B). Analysis of IL-13ᵉᴳᶠᴾ⁺ and IL-17ᵉʸᶠᴾ⁺ cells revealed that CD4⁺ T cell depletion substantially reduced both IL-13 and IL-17 cytokine expressing cells in the BAL fluid, lung tissue and dLNs (Fig. 2B and Supplementary Fig. 5C). Moreover, analysis of secreted factors in the BAL fluid identified that CD4⁺ T cell depleted mice showed reduced levels of both type 2 and type 17 inflammatory mediators (Supplementary Fig. 5D). Finally, CD4⁺ T cell depletion resulted in reduced lung inflammatory pathology, as measured by histology (Supplementary Fig. 5E).

We next assessed the importance of γδ T cells, the other major source of IL-17 in our analyses (Fig. 1E and Supplementary Fig. 3C), in the pulmonary response to *Af* spores, comparing γδ T cell deficient (*Tcrd*⁻/⁻) to WT mice. The influx of granulocytes and lymphoid populations to the airways or lung tissue of *Af*-exposed *Tcrd*⁻/⁻ mice was similar (or even elevated) to WT controls (Fig. 2C and Supplementary Figs. 5E, F), as were levels of inflammatory mediators in the BAL fluid (Supplementary Fig. 5G). Moreover, the proportions of type 2 and type 17 cytokine expressing CD4⁺ T cells in the lung tissue were increased in *Af*-exposed γδ T cell deficient mice (Fig. 2D).

Together, these data demonstrate that CD4⁺ T cells are critical for development of pulmonary type 2 and type 17 inflammatory responses against *Af*, while IL-17 secreting γδ T cells are not essential for this process.

### Priming of pulmonary fungal type 2 and type 17 allergic inflammation is dependent on CD11c⁺ cells

After identifying that CD4⁺ T cells were critical for allergic airway inflammation against *Af* spores, we next wanted to understand which APCs were responsible for initiating and directing this response. We used transgenic *Cd11c.DOG* mice, which have been employed extensively to demonstrate the role of CD11c⁺ APCs in the induction of inflammatory responses against HDM and helminths[18,41], to deplete CD11c⁺ DCs and alveolar macrophages (AlvMΦs) and assess if this altered the onset of type 2 and/or type 17 anti-*Af* allergic airway inflammation (Fig. 3A). Administration of diphtheria toxin (DTx) to *Cd11c.DOG* mice during spore exposure significantly depleted CD11c-expressing AlvMΦs and PDCA1⁺ plasmacytoid DCs (pDCs) in the lung (Fig. 3A, Supplementary Fig. 6A, B). As expected, interstitial MΦs (IntMΦs) were not significantly affected by DTx treatment (Fig. 3A and Supplementary Fig. 5A). Unexpectedly, lung tissue MHCII⁺CD11c⁺ conventional DCs (cDCs) were also not significantly depleted following DTx administration (Fig. 3A). More refined analysis of cDC subsets showed effective depletion of MHCII⁺CD11c⁺XCR1⁺ cDC1s, but not MHCII⁺CD11c⁺CD11b⁺ cDC2s, following DTx treatment (Fig. 3B). CD301b (Mgl2, MΦ Gal/GalNAc lectin-2) has previously been identified as a marker of a subset of cDC2s with type 2 and 17 polarising capabilities[42,43], and CD64 (FcγR4) as a marker of 'inflammatory' cDC2s that are dependent on monocyte infiltration (Inf DCs)[44]. Use of these markers to further dissect the impact of CD11c depletion on lung cDC2 subsets revealed significantly reduced Mgl2⁺ cDC2s, with a concurrent increase in CD64⁺ Inf DCs in DTx treated *Cd11c.DOG* mice (Fig. 3B). Thus, these data show that administration of DTx to *Cd11c.DOG* mice during *Af* spore exposure effectively depleted lung pDCs, cDC1s, Mgl2⁺cDC2s and AlvMΦs, while leaving IntMΦs unaffected and boosting Inf DCs.

Depletion of CD11c⁺ cell populations during repeat spore exposure resulted in significantly reduced eosinophilia in the BAL fluid and lung tissue, while neutrophilia was increased (Fig. 3C and Supplementary Fig. 5B), in line with previously reported systemic neutrophilia (including in the lung) in DTx treated *Cd11c.DOG* mice[45]. Although CD11c depletion did not reduce lymphoid cell numbers in the BAL fluid or lung tissue (Supplementary Fig. 6C), pulmonary CD4⁺ T cell cytokine profiles were dramatically altered, with a significant decrease in production of both type 2 and type 17 cytokines (Fig. 3D). This was accompanied with a reduction in levels of allergic inflammatory markers and IL-12p40 in the BAL fluid, and lung inflammatory pathology

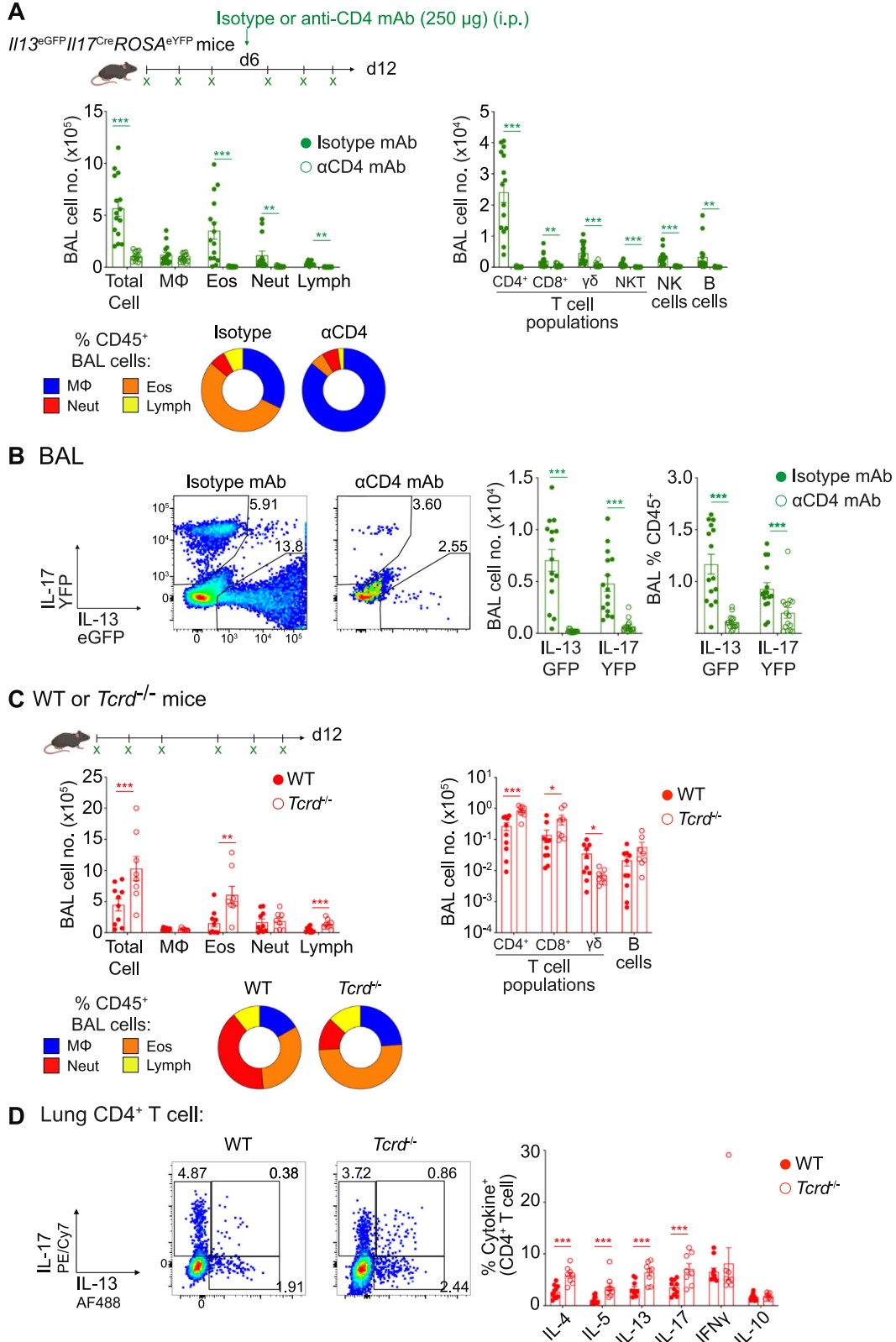

(Supplementary Fig. 6D, E). Together, these data show that depletion of CD11c⁺ cells (including pDCs, cDC1s, CD301b⁺cDC2s and AlvMΦs) reduces eosinophilia, and both type 2 and type 17 cytokine production by CD4⁺ T cells, in the pulmonary response to *Af* spores.

As pDCs were depleted in DTx treated *Cd11c.DOG* mice, and given previous work that has proposed that pDC subsets may have a role in anti-fungal and/or allergic inflammation[22,23], we used *Bdca2*[DTr] mice to

selectively deplete pDCs[46] to see if this impacted fungal allergic airway inflammation. Treatment of *Bdca2*[DTr] mice with DTx significantly reduced proportions and numbers of pDCs, but had no measurable impact on cDC or MΦ populations in the lung tissue or BAL fluid (Supplementary Fig. 7A, B). Despite dramatically reduced pDCs, BAL fluid eosinophilia and neutrophilia was not altered following repeat *Af* spore exposure (Supplementary Fig. 7C).

**Fig. 2 | Type 2 and type 17 fungal allergic airway inflammation is dependent on CD4⁺ T cells. A** $Il13$^eGFP^$Il17$^Cre^$ROSA$^eYFP^ mice were treated with anti-CD4 mAb or isotype control mAb after three doses of $Af$ spores ($4 \times 10^5$ per dose) and were subsequently exposed to three further doses of spores. Created in BioRender. Cook, P. (2024) https://BioRender.com/z24b189. Graphs display the number and percentage of macrophages (MΦs), eosinophils (Eos), neutrophils (Neut) and lymphocytes (Lymph) or the number of lymphocyte populations isolated detected in the BAL fluid 24 h after the sixth dose. **B** Flow cytometry plots and graphs identify the proportion and number of $Il13$ and $Il17$ expressing cells isolated from the BAL fluid. **C** WT or $Tcrd$^-/-^ mice were exposed to six doses of $Af$ spores ($4 \times 10^5$ per dose). Created in BioRender. Cook, P. (2024) https://BioRender.com/z24b189. Graphs display the number and percentage of MΦs, eosinophils (Eos), neutrophils (Neut) and lymphocytes (Lymph) or the number of lymphocyte populations detected in the BAL fluid 24 h after the sixth dose. **D** Representative flow cytometry plots identify IL-13⁺ and IL-17⁺ populations via intracellular cytokine staining of lung CD4⁺ T cells post stimulation with PMA/ionomycin. Graphs show the proportion of CD4⁺ T cells expressing type 1, type 2 and type 17 cytokine. **A & B** data are from 3 independent experiments ($n = 30$ biologically independent animals). **C & D** data are from 2 independent experiments ($n = 18$ biologically independent animals). Data were fit to a linear mixed effect model, with experimental day as a random effect variable, and groups compared with a two-sided Tukey's multiple comparison test. *$p < 0.05$, **$p < 0.01$, ***$p < 0.001$. Data are presented as mean values ± SEM. **B** $P$ Values, Total Cell; anti-CD4 mAb vs isotype control mAb = $5.53 \times 10^{-11}$. Eos; anti-CD4 mAb vs isotype control mAb = $9.95 \times 10^{-7}$. Neut; anti-CD4 mAb vs isotype control mAb = 0.00232. Lymph: anti-CD4 mAb vs isotype control mAb = $4.82 \times 10^{-12}$. CD4⁺ T cells; anti-CD4 mAb vs isotype control mAb = $5.79 \times 10^{-13}$. CD8 + T cells; anti-CD4 mAb vs isotype control mAb = 0.00407. γδ T cells; anti-CD4 mAb vs isotype control mAb = $1.11 \times 10^{-6}$. NK T cells; anti-CD4 mAb vs isotype control mAb = $5.85 \times 10^{-5}$. NK cells; anti-CD4 mAb vs isotype control mAb = $1.44 \times 10^{-6}$. B cells; anti-CD4 mAb vs isotype control mAb = 0.00557. C, $P$ Values, IL-13eGFP+ cell number; anti-CD4 mAb vs isotype control mAb = $5.18 \times 10^{-11}$. IL-17 YFP⁺ cell number; anti-CD4 mAb vs isotype control mAb = $2.01 \times 10^{-7}$. % IL-13eGFP⁺; anti-CD4 mAb vs isotype control mAb = $9.68 \times 10^{-11}$. IL-17 YFP⁺ cell number; anti-CD4 mAb vs isotype control mAb = 0.00047. **D** $P$ values, Total cell; WT vs $Tcrd$–/– = 0.000662. Eos; WT vs $Tcrd$^-/-^ = 0.00115. Lymph; WT vs $Tcrd$^-/-^ = 0.00024. CD4⁺ T cells; WT vs $Tcrd$^-/-^ = 4.42 x $10^{-6}$. CD8⁺ T cells; WT vs $Tcrd$^-/-^ = 0.0122. γδ T cells; WT vs $Tcrd$^-/-^ = 0.0214. **E** $P$ values, IL-4; WT vs $Tcrd$^-/-^ = $3.31 \times 10^{-6}$. IL-5; WT vs $Tcrd$^-/-^ = 0.000117. IL-13; WT vs $Tcrd$^-/-^ = $3.88 \times 10^{-8}$. IL17; WT vs $Tcrd$^-/-^ = 0.000758. Source data are provided as a Source Data File. Schematics in figures were created in https://BioRender.com.

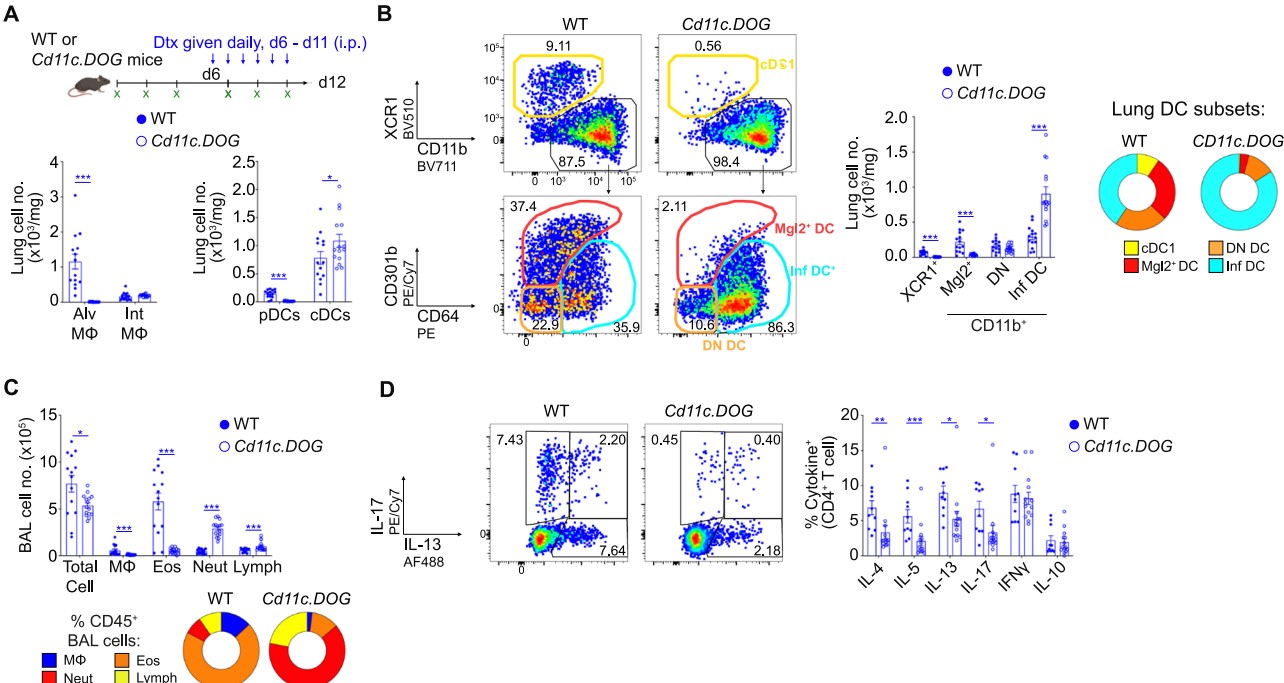

**Fig. 3 | Type 2 and type 17 fungal allergic airway inflammation is dependent on CD11c-expressing cells. A** WT or $Cd11c.DOG$ mice were treated with diphtheria toxin (DTx) after three doses of $Af$ spores ($4 \times 10^5$ per dose) and this continued daily as mice were subsequently exposed to three further doses of spores. Created in BioRender. Cook, P. (2024) https://BioRender.com/z24b189. Graphs show the number of alveolar macrophages (AlvMΦs), interstitial macrophages (IntMΦs), plasmacytoid DCs (pDCs) and DC subsets detected in the lung tissue 24 h after the sixth dose. **B** Flow cytometry plots identifying DC subsets alongside graphs showing the number amd proportion of these populations detected in the lung tissue. **C** Graphs show the number and percentage of macrophages (MΦs), eosinophils (Eos), neutrophils (Neut) and lymphocytes (Lymph) detected in the BAL fluid 24 h after the sixth dose. **D** Representative flow cytometry plots identify IL-13⁺ and IL-17⁺ populations via intracellular cytokine staining of lung CD4⁺ T cells post stimulation with PMA/ionomycin. Graphs show the proportion of CD4⁺ T cells expressing type 1, type 2 and type 17 cytokine. **A–C** data are from 3 independent experiments ($n = 29$ biologically independent animals). **D** data are from 3 independent experiments ($n = 24$ biologically independent animals). Data were fit to a linear mixed effect model, with experimental day as a random effect variable, and groups compared with a two-sided Tukey's multiple comparison test. *$p < 0.05$, **$p < 0.01$, ***$p < 0.001$. Data are presented as mean values ± SEM. **A** $P$ Values, Alv MΦ; WT vs $Cd11c.DOG$ = $7.68 \times 10^{-11}$. pDCs; WT vs $Cd11c.DOG$ = < $2.0 \times 10^{-16}$. cDC; WT vs $Cd11c.DOG$ = 0.0156. **B** $P$ Values, XCR1⁺; WT vs $Cd11c.DOG$ = $2.44 \times 10^{-15}$. Mgl2⁺; WT vs $Cd11c.DOG$ = $5.4 \times 10^{-7}$. Inf DC; WT vs $Cd11c.DOG$ = $2.82 \times 10^{-10}$. **C** $P$ Values, Total Cell; WT vs $Cd11c.DOG$ = 0.0104. MΦ; WT vs $Cd11c.DOG$ = 0.000902. Eos; WT vs $Cd11c.DOG$ = $1.31 \times 10^{-9}$. Neut; WT vs $Cd11c.DOG$ = < $2.0 \times 10^{-16}$. Lymph; WT vs $Cd11c.DOG$ = 0.000517. **D** $P$ Values, IL-4; WT vs $Cd11c.DOG$ = 0.00666. IL-5; WT vs $Cd11c.DOG$ = 0.000783. IL-13; WT vs $Cd11c.DOG$ = 0.0138. IL-17; WT vs $Cd11c.DOG$ = 0.0119. Source data are provided as a Source Data File. Schematics in figures were created in https://BioRender.com.

Furthermore, CD4⁺ T cell type 2 and type 17 cytokines were similar in $Bdca2$^DTr^ and WT control mice (Supplementary Fig. 7C), demonstrating that pDCs are not essential for promotion of anti-$Af$ allergic airway inflammation.

## Single cell RNA-sequencing defines pulmonary DC heterogeneity during fungal allergic airway inflammation
Having established that CD11c⁺ cells, but not pDCs, were required for generation of airway inflammation against $Af$ spores, we took a less

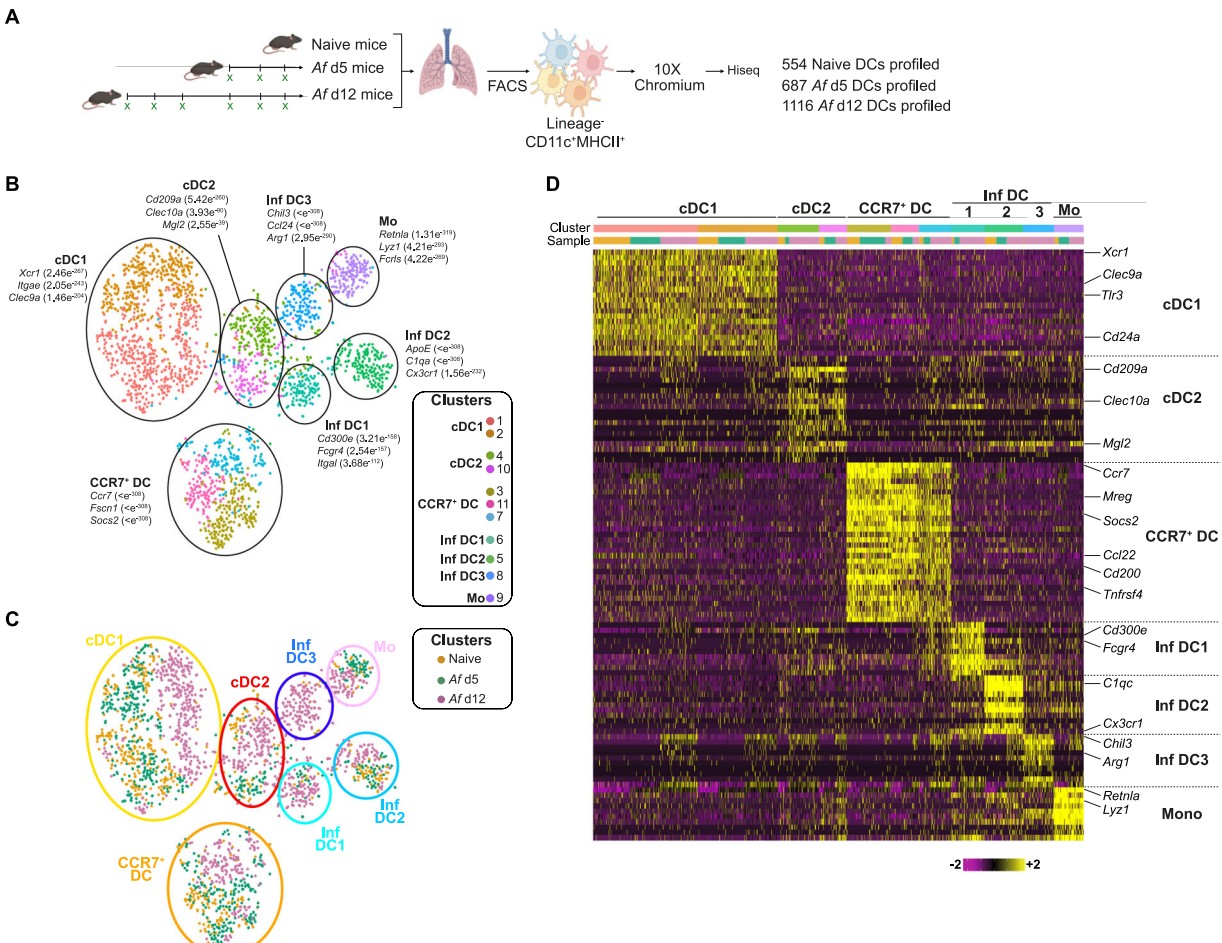

**Fig. 4 | Single cell RNA-sequencing of pulmonary DCs reveals distinct clusters that can be identified throughout fungal allergic airway inflammation. A** DCs were isolated from the lung tissue of naïve mice or mice repeatedly exposed intransally to *Af* spores ($4 \times 10^5$ per dose), harvested 24 h after the third and sixth dose (*Af* d5 and *Af* d12, respectively). Single cell libraries were generated from these populations using the 10x chromium platform and sequenced (Illumina Hi-seq). Created in BioRender. Cook, P. (2024) https://BioRender.com/n00n566. **B** tSNE of all cells coloured according to the clusters. 11 clusters, each defined by the colours indicated in the key, were identified by dynamic tree cluster method using highly expressed genes. These were refined to 7 distinct clusters of DCs (circled) based on highly expressed genes as indicated. **C** tSNE plots indicate the distribution of cells within each cluster based on their origin (orange: naïve, green: *Af* d5 and purple: *Af* d12). **D** Heat map shows scaled gene expression of top 10 defining genes for each cluster. Schematics in figures were created in https://BioRender.com.

subjective approach to identify changes in other CD11c⁺ DC populations in this setting. While cDC subsets can be cleanly identified in most tissues in the steady state, this can be much more challenging during inflammation, where they are likely a heterogenous mixture of different sub-populations. For example, previous studies have suggested that moDCs (which are hard to distinguish from cDC2s by flow cytometry) may elicit allergic inflammatory responses[19]. The application of single cell technologies, particularly scRNAseq, has dramatically improved our understanding of DC subset heterogeneity[20], including revealing the possibility of a third DC subset (DC3) which expresses high levels of CCR7[47], and Inf DCs that arise from cDC1s or cDC2s, rather than monocytes, during sustained inflammation[44]. However, the impact of fungi on DC subsets during allergic airway inflammation is poorly understood, as is the identity of the subset(s) responsible for promoting type 2 versus type 17 fungal allergic airway responses.

To define DC subset heterogeneity during development of allergic airway inflammation against *Af* spores, we employed scRNAseq of CD11c⁺MHCII⁺ DCs FACS isolated from the lungs of WT mice that were unchallenged (naïve) (554 cells) or had been exposed to three doses (d5, prior to induction of allergic inflammation, 687 cells) or six doses (d12, developed allergic responses, 1116 cells) of *Af* spores (Fig. 4A). Unbiased analysis (dynamic-tree clustering) revealed 11 clusters which, based on cluster defining gene expression, were further refined to 7

possible subsets (Fig. 4B, C). To decipher the identify of these clusters, analysis of common cDC markers (*Dpp4*, *Zbtb46*, *Flt3*) versus markers associated with monocytes and Inf DCs (*Axl*, *Csf1r*, *Ccr2*, *Cd68*, *Fcgr1*, *Mafb*) clearly separated putative cDC versus Inf DC subsets (Supplementary Fig. 8A). Analysis of transcripts that were significant in defining clusters clearly identified cDC1s, with expression of genes that are essential for cDC1 development, function and phenotype (e.g. *Clec9a*, *Irf8*, *Itgae*) (Fig. 4B–D, and Supplementary Fig. 8B). In addition, a cDC2 cluster was identified, with significant expression of transcripts previously associated with this subset (e.g. *Cd209a*, *Clec10a* that encodes *CD301a* and *Mgl2* that encodes *CD301b*) (Fig. 4B–D and Supplementary Fig. 9A). A third cDC cluster, distinct from either cDC1s or cDC2s based on a range of transcripts, expressed high levels of *Ccr7*. This CCR7⁺ DC cluster, along with expression of several transcripts (e.g. *Mreg*, *Socs2*, *Cd200*), appeared to have a similar gene profile to a population previously called 'Mreg' DCs[47] (Fig. 4D and Supplementary Fig. 9B), which have been proposed to boost Treg populations during cancer[47]. Strikingly, these three cDC clusters were present in cells isolated from naïve or d5 or d12 *Af*-exposed mice (Fig. 4B, C). Together, these data suggest that the cDC1, cDC2 and CCR7⁺ clusters we identified by scRNAseq represent separate DC subsets that can be found in the lung both at steady state and during fungal allergic airway inflammation.

Of the remaining four clusters, which expressed transcripts associated with monocytes and Inf DCs, one appeared to be closely related to undifferentiated recruited monocytes (Mo), with significant expression of *Retnla* and *Lyz1* (Fig. 4B–D and Supplementary Fig. 10A), suggestive of monocytes that may have recently up-regulated CD11c and MHCII[48,49]. Additionally, we identified three separate clusters of DCs which we labelled as Inf DCs based on their gene signatures. The first of these clusters (Inf DC1) expressed genes previously associated with monocyte and DC populations (e.g. *Cd300e*, *Fcgr4*, *Itgal* and *Fcer1g*) (Fig. 4B–D and Supplementary Fig. 10B). Cluster 2 (Inf DC2) expressed a range of genes associated with monocyte derived cells (e.g. *C1qc*, *Trem2*) (Fig. 4D and Supplementary Fig. 10C). Notably, both Inf DC1 and Inf DC2 expressed high levels of *Cx3cr1*, a chemokine receptor that is well known to be highly expressed by MΦs, DCs and circulating monocytes[50]. Although these three clusters (Mo, Inf DC1 and Inf DC2) were present in cells isolated from both naïve and *Af*-exposed mice, more cells in each cluster were from mice that had been exposed to *Af* (d5 and d12) (Fig. 4C). In contrast, the remaining cluster (Inf DC3) were mainly evident at later stages of anti-*Af* allergic airway inflammation (d12) (Fig. 4C). Furthermore, cells in this cluster were found to express high levels of *Chil3*, *Ccl24* and *Arg1*, genes associated with MΦs responding to type 2 cytokines[51] (Supplementary Fig. 10D). Finally, one of the genes identified within in the cDC2 cluster (*Mgl2*) was also expressed on some cells within the Inf DC2 and Inf DC3 clusters, albeit at a relatively lower level. Together, this analysis revealed several subsets of Inf DCs which, in addition to cDC1s, cDC2s and CCR7+ DCs, could contribute to generation of type 2 or type 17 aspects of allergic airway inflammation against *Af*.

Although DCs are essential for induction of allergic airway inflammation against *Af* (Fig. 3) or HDM challenge[18,19], the mechanism(s) employed by DCs to induce these responses are poorly understood. To address this, we assessed whether the distinct DC subsets we had identified by scRNAseq expressed candidate genes that could relate to their ability to promote either type 2 or type 17 features of allergic airway inflammation. Surprisingly, expression of *Irf4*, essential for cDC2 development and induction of type 2 and type 17 inflammation[32,52], was not restricted to any particular DC cluster (Supplementary Fig. 11A). Similarly, genes previously suggested to be associated with the ability of DCs to promote type 2 inflammation (e.g. *H2.Eb2*, *Jag1*, *Mbd2*, *Raspgrp3*, *Stat5a*, *Stat5b*[53–56]) were not restricted to a specific DC cluster (Supplementary Fig. 11A). Although distinct co-stimulatory molecules and cytokine receptors have been associated with DC promotion of type 2 and type 17 inflammation (e.g. *Cd40*, *Cd86*, *Il1rl1*, *Il4ra*, *Il13ra1*, *Pdcd1lg2*)[52,57,58], expression of these was also not restricted to a particular DC cluster (Supplementary Fig. 11A, B). However, expression of *Pdcd1lg2* (PDL2, DC expression of which has been proposed to promote type 2 responses[52]) was also prominent in cDC2s, CCR7+ DCs and Inf DC3s (Supplementary Fig. 10A). Analysis of cytokine genes across the clusters revealed that *Il1b*, implicated in Th2 and Th17 CD4+ T cell proliferation[59,60], was particularly expressed by cDC2s and, to some extent, Inf DCs (Supplementary Fig. 10C). In contrast to cDC2s and Inf DCs, CCR7+ DCs were the main expressors of *Il12b* and *Il15*, cytokines linked to type 1 and type 17 immunity[57] (Supplementary Fig. 11C). Conversely, expression of *Il6* (a cytokine with broad inflammatory impact and detectable in the BAL fluid) was highly expressed by the Mo cluster. Chemokines have also been implicated in allergic airway inflammation and several were highly expressed in certain DC clusters (Supplementary Fig. 11D). The majority of chemokines were identified in the CCR7+ DC and Inf DC clusters, whereas the cDC2 cluster generally expressed low levels of chemokines. However, along with several Inf DC clusters, cDC2s expressed *Ccl9* which has been implicated in DC recruitment[61] but is poorly studied in allergic disease. The CCR7+ DC cluster expressed *Ccl5*, *Ccl17* (along with cDC1s and Inf DC3) and *Ccl22*, mediators we have previously identified as features of type 2 polarising DCs[54]. Inf DC2 and Mo expressed a range

of genes associated with inflammatory cell recruitment, especially Cxcl2, which has shown to be important for AlvMΦ anti-fungal immunity and neutrophil recruitment[62]. Furthermore, Inf DC3s appeared to express *Ccl24* which, in addition to *Ccl17*, is a factor associated with DC ability to induce allergic airway inflammation[63]. Interestingly, we observed no clear pattern of altered expression of these functionally defining genes in each cluster over the course of *Af* exposure. To examine this further, we undertook gene-set enrichment analysis of these data and discovered that genes associated with metabolism (especially oxidative phosphorylation), antigen processing and phagosome activity were upregulated across cDC1s, CCR7+ DCs, cDC2s, Inf DC2s, Inf DC3s and Mos from *Af*-exposed mice (particularly at d12, after 6 doses) (Supplementary Fig. 12A, B). In summary, use of scRNAseq has revealed that particularly cDC1s, cDC2s, CCR7+ DCs, Inf DC2s and Inf DC3s express a range of genes that have previously been associated with DC induction of type 2 and/or type 17 responses, implicating these DC clusters as the likeliest candidates for promoting fungal allergic airway inflammation.

## Confirmation of pulmonary DC subset heterogeneity at the protein level during fungal allergic airway inflammation

To verify whether the DC subsets we had identified in lung tissue collected from *Af*-exposed mice by mRNA expression were also evident at a protein level, we used key genes associated with each DC cluster (Fig. 4) to develop a 36-parameter antibody panel for mass cytometry (CyTOF) (Supplementary Figs. 13A and 14A). Unbiased (t-SNE and FlowSOM[64]) analysis of Ab staining using 21 cluster defining markers enabled us to designate each DC subset based on protein expression (Supplementary Fig. 14). In agreement with previous studies[44,65] and our scRNAseq data (Supplementary Fig. 8), we found expression of CD26 (*Dpp4*) reliably separated the cDC (cDC1, cDC2 and CCR7+ DCs) from Inf DC / Mo clusters (Supplementary Fig. 12). As expected, cDC1s could be clearly identified as CD26hiXCR1+CD11bloSIRPα−, while CD11b and SIRPα showed sufficient overlap in expression that either marker could be used to separate cDC1s from other subsets (Supplementary Fig. 14). Three clusters of cDC2s (CD11b+SIRPα+CD26hi) were identified based on Mgl2 and CD209a (C-type lectin receptor SIGNR5 and homologue of human DC-SIGN) expression, in keeping with our scRNAseq data showing cDC2 cluster cells as Mgl2+ and/or CD209a+ (Fig. 4). From the scRNAseq defined CCR7+ DCs (Fig. 4), Abs staining CD200 and CD215 was not useful as a cluster defining marker (Supplementary Fig. 14B). As we were unable to obtain reproducible Ab staining of CCR7, we instead identified CCR7+ cells via use of labelled CCL19[66], which gave clear staining profiles in a number of DC subsets (Supplementary Fig. 14B). Indeed, several clusters contained a mixture of CCL19lo and CCL19hi cells, the latter likely representing DCs that highly express CCR7. Furthermore, along with CD63 and CD134, this CCL19 staining allowed us to identify a CCR7+ DC cluster (CD11b+SIRPα+CD26+CD63+CD134loCCL19+) in line with our scRNAseq data (Supplementary Fig. 14). The three Inf DC clusters were first defined as CD11b+SIRPα+CD26loCD64+, then further separated into Inf DC1s (CX3CR1+FcγR4hi), Inf DC2s (Mgl2−/loCX3CR1+) and Inf DC3s (Mgl2loYm1+). Finally, two separate Mo populations were identified based on expression of CD64 and Ly6C (Mono 1: Ly6C+, Mono 2: Ly6C−). A cluster of SIRPα+CD11b+CD26+ DCs that were not expressing any of the other markers was identified, which we termed as CD11b+ DCs. In addition to CD200 and CD215, RELMα, FcεR1 and Arg1 were not effective at clustering DCs. Thus, use of CyTOF enabled us to verify the pulmonary DC subsets suggested by our scRNAseq data (Fig. 4). However, we experienced significant sample loss during CyTOF acquisition, so developed a revised panel for flow cytometry, building on the markers we had identified to be effective using CyTOF. This informed flow cytometry approach also successfully distinguished the pulmonary DC subsets we had identified by scRNAseq (Fig. 5A, Supplementary Figs. 13B and 15). Together, these complementary CyTOF

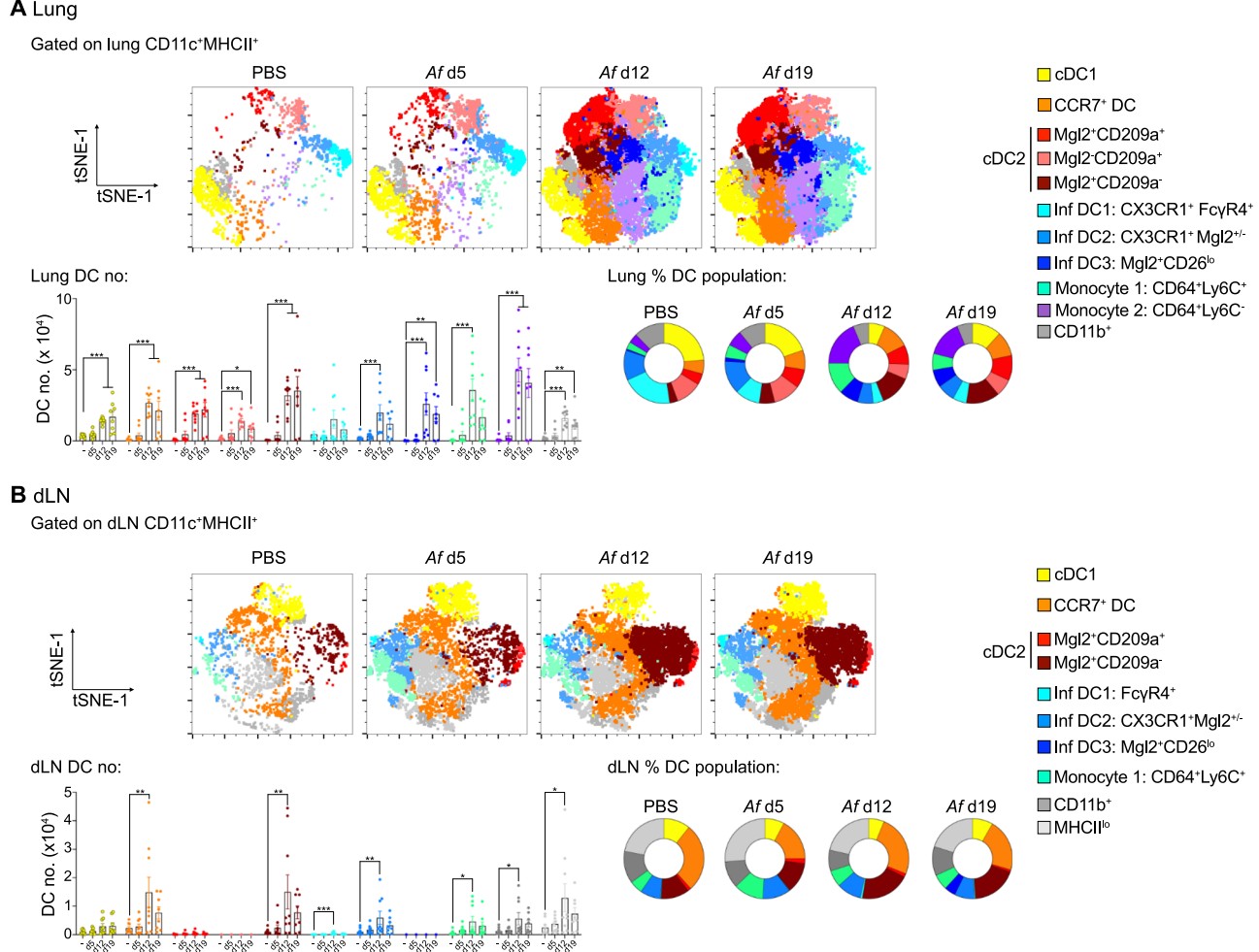

**Fig. 5 | Repeat fungal exposure increases the proportion of cDC2s in the lung and dLN.** Representative tSNE plots reveal DC subsets identified by unbiased clustering analysis of flow cytometry data from the (**A**) lung and (**B**) dLNs of mice repeatedly exposed intranasally to PBS or *Af* spores (4 × 10⁵ per dose). Graphs display the numbers and percentage of DC subsets isolated from the (**A**) lung and (**B**) dLNs. Data are from 2 independent experiments (*n* = 35 biologically independent animals). Data were fit to a linear mixed effect model, with experimental day as a random effect variable, and groups compared with a two-sided Tukey's multiple comparison test. *$p < 0.05$, **$p < 0.01$, ***$p < 0.001$. Data are presented as mean values ± SEM. **A** *P* Values, cDC1; PBS vs *Af* d12 = 3.53 × 10⁻⁷, PBS vs *Af* d19 = 0.000142. CCR7⁺ DC; PBS vs *Af* d12 = 5.422 x 10⁻⁷, PBS vs *Af* d19 = 0.000142.

cDC2 Mgl2⁺CD209a⁺; PBS vs *Af* d12 = 2.07 × 10⁻⁵, PBS vs *Af* d19 = 1.35 10⁻⁶. cDC2 Mgl2⁻CD209a⁺; PBS vs *Af* d12 = 1.08 × 10⁻⁵, PBS vs *Af* d19 = 0.0139. cDC2 Mgl2⁺CD209a⁻; PBS vs *Af* d12 = 4.05 × 10⁻⁵, PBS vs *Af* d19 = 1.02 × 10⁻⁵. Inf DC2; PBS vs *Af* d12 = 4.32 × 10⁻⁵. Inf DC3; PBS vs *Af* d12 = 7.25 × 10⁻⁵, PBS vs *Af* d19 = 0.00422. Monocyte 1; PBS vs *Af* d12 = 3.02 × 10⁻⁶. Monocyte 2; PBS vs *Af* d12 = 2.29 × 10⁻⁸, PBS vs *Af* d19 = 1.89 × 10⁻⁵. CD11b⁺; PBS vs *Af* d12 = 6.57 × 10⁻⁷, PBS vs *Af* d19 = 0.00209. **B** *P* Values, CCR7⁺ DC; PBS vs *Af* d12 = 0.00668. cDC2 Mgl2⁺CD209a⁺; PBS vs *Af* d12 = 2.07 × 10⁻⁵, PBS vs *Af* d19 = 1.35 × 10⁻⁶. cDC2 Mgl2⁺CD209a⁻; PBS vs *Af* d12 = 0.00512. Inf DC1; PBS vs *Af* d12 = 0.000319. Inf DC2; PBS vs *Af* d12 = 0.00841. Monocyte 1; PBS vs *Af* d12 = 0.0234. CD11b⁺; PBS vs *Af* d12 = 0.023. MHCIIˡᵒ; PBS vs *Af* d12 = 0.014. Source data are provided as a Source Data File.

and flow cytometry approaches confirm that the main pulmonary DC subsets suggested by our scRNAseq mRNA expression data exist at a protein level.

Having developed an effective flow cytometry approach to identify pulmonary DC subsets, we could then track DC population heterogeneity and dynamics during *Af*-induced allergic airway inflammation. The proportion and number of DCs increased in the lung by d12, and were maintained with further *Af* exposure (Supplementary Fig. 13C, D), in keeping with the time points where allergic airway inflammation became apparent (Fig. 1). At steady state, the majority of lung DCs were either cDC1s or cDC2s (consisting of Mgl2⁺CD209a⁺ or Mgl2⁻CD209a⁺ cDC2s), along with Inf DC1s and a minor population of CCR7⁺ DCs (Figs. 5A and Supplementary Fig. 15). After initial exposure to *Af* spores (d5), prior to overt development of allergic airway inflammation, DC subsets were largely unchanged compared to naïve mice. However, further exposure to spores (d12 and d19) led to a clear increase in the proportion of cDC2s (especially Mgl2⁺CD209a⁻ cDC2s), CCR7⁺ DCs and monocytes, while cDC1s and Inf

DC1s decreased (Fig. 5A and Supplementary Fig. 15). However, due to the increase in lung DCs following 6 and 9 spore doses (Supplementary Fig. 13D), the total number of most DC populations (except Inf DC1s) increased in the lungs (Fig. 5A). Overall, these data show that there is a dramatic alteration in lung tissue DC heterogeneity as fungal allergic airway inflammation progresses, with significantly expanded cDC2s and CCR7⁺ DCs.

An important step in DC orchestration of allergic airway inflammation is the migration of antigen exposed DCs to the dLNs to activate CD4⁺ T cells. Using our DC subset flow cytometry panel, we could identify a trend for overall increased DC numbers in the lung dLNs coincident with the onset of allergic airway inflammation (Fig. 5B and Supplementary Fig. 13D). Unbiased analysis of dLN DC populations showed that fewer subsets were represented in the dLNs compared to the lung, suggesting that not all of the lung DC subsets were migrating from the tissue to the dLNs. The most dramatic change seen in the dLNs as allergic airway inflammation developed was an increase in the proportion and number of Mgl2⁺CD209a⁻ cDC2s (Fig. 5B and

Supplementary Fig. 16). While several cDC2 subsets were apparent in the lung, very few CD209a⁺ cDC2s reached the dLNs. Despite being one of the smaller subsets in the lung tissue, one of the major clusters identified in the dLNs was CCR7⁺ DCs (~ 30% of DCs) which increased in number upon allergic airway inflammation. Previous studies have shown that high expression of MHCII can separate migratory versus resident dLN DC populations[67]. Analysis of MHCII expression showed that cDC2s and CCR7⁺ DCs accounted for the bulk of MHCII^hi cells in the dLNs, suggesting that these subsets form the majority of migratory DCs (Supplementary Fig. 16). Although lung DCs included several Inf DC subsets that expanded during allergic airway inflammation (Fig. 5A), comparatively few Inf DCs were detected in the dLNs, and they did not dramatically increase following *Af* exposure (Fig. 5B and Supplementary Fig. 16). Similarly, allergic airway inflammation had little impact on dLN cDC1s and monocytes. In summary, we have used scRNAseq to develop a high throughput flow cytometry based approach to identify DC populations at a protein level, and in doing so revealed that Mgl2⁺ cDC2s are the major subset which increases in the lung tissue and dLNs during development of allergic airway inflammation against *Af*.

### Mgl2⁺ cDC2s promote type 2, but not type 17, responses during fungal allergic airway inflammation

While one of the clearest changes in the lung and dLNs following repeat *Af* exposure was an increase in Mgl2⁺ cDC2s, it was unclear whether these were essential for promoting fungal allergic airway inflammation. To test this, we used *Mgl2*-DTR mice to deplete Mgl2-expressing cells (Fig. 6A), an approach that has previously been shown to reduce type 2 responses against helminth infection and impair type 17 responses against bacterial infection[43,68]. Previous studies have highlighted that Mgl2 is expressed by lung MΦ populations, especially IntMΦs[69]. However, DTx treatment of *Af*-exposed *Mgl2*-DTR^het mice did not reduce the number of lung AlvMΦs and IntMΦs compared to *Af*-exposed WT (Fig. 6B). In the absence of spores, we observed that IntMΦs (but not AlvMΦs) were slightly reduced in *Mgl2*-DTR^het mice treated with DTx (Supplementary Fig. 17A). In contrast, DTx treatment dramatically reduced the number of Mgl2⁺ cDC2s (Mgl2⁺CD209a⁺ and Mgl2⁺CD209a⁻) in the lungs of *Mgl2*-DTR^het mice in the absence of spores (Supplementary Fig. 17B). After six doses of spores, DTx treatment reduced the number of lung DCs (Supplementary Fig. 17C), caused by a striking reduction in the number and proportion of Mgl2⁺ cDC2s (Mgl2⁺CD209a⁺ and Mgl2⁺CD209a⁻) in *Af*-exposed *Mgl2*-DTR^het mice (Fig. 6C and Supplementary Fig. 17D). DTx administration also significantly reduced CCR7⁺ DCs, Mgl2⁻CD209a⁺ cDC2s, Inf DC2s and Inf DC3s in the lung. Analysis of the dLNs showed a similar dramatic reduction of Mgl2⁺ cDC2s (Mgl2⁺CD209a⁺ and Mgl2⁺CD209a⁻) but depletion did not alter the number of the other DC subsets in dLNs recovered from *Af*-exposed *Mgl2*-DTR^het mice (Fig. 6C and Supplementary Fig. 17E). Overall, these data showed that DTx treatment of *Mgl2*-DTR^het mice affected several DC subsets in the lungs and dLNs during fungal allergic airway inflammation, but that cDC2s were the most dramatically depleted in both sites (especially in the dLNs).

To determine the impact of Mgl2⁺ cell depletion on anti-*Af* allergic airway responses, we analysed BAL fluid and lung tissue isolated from spore exposed, DTx treated, *Mgl2*-DTR^het and WT mice. Pulmonary eosinophilia was strikingly and significantly reduced following Mgl2⁺ cell depletion, whereas neutrophilia was largely maintained (similar proportions, with slightly reduced cell number) and other cell types were unaltered (e.g. MΦs) in the BAL fluid (Fig. 6E and Supplementary Fig. 17F, G). Despite having identified that allergic inflammation was dependent on CD4⁺ T cells (Fig. 2), we found that DTx administration had no impact on the number of CD4⁺ T cells in BAL fluid and lung tissue in *Af*-exposed *Mgl2*-DTR^het mice (Supplementary Fig. 17H). However, analysis of CD4⁺ T cell cytokine production in the lung in *Mgl2*-DTR^het mice showed that their potential to produce type 2, but

not type 17, cytokines was significantly reduced following DTx treatment (Fig. 6F). Analysis of the BAL fluid and histology showed that DTx administration significantly reduced levels of allergic inflammatory mediators, and inflammatory pathology, in *Af*-exposed *Mgl2*-DTR^het mice (Supplementary Fig. 17I, J). This alteration in allergic inflammation had no impact on spore clearance, as DTx administration to *Mgl2*-DTR^het mice had no measurable impact on lung fungal burden (Supplementary Fig. 17K).

Although cDC1s were not significantly affected by DTx treatment of *Mgl2*-DTR^het mice, they did upregulate expression of genes associated with metabolism and antigen processing (Fig. 4 and Supplementary Fig. 12). To directly address whether they were involved in promoting type 2 or type 17 anti allergic inflammation, we utilised *Batf3*⁻/⁻ mice to see if cDC1 deficiency[70] impacted fungal allergic airway inflammation. Analysis of *Batf3*⁻/⁻ mice following *Af* spore exposure showed clear lung cDC1 deficiency, with other DC populations remaining in-tact (Supplementary Fig. 18A). Despite cDC1 deficiency, eosinophilia and neutrophilia in the BAL fluid was not altered in *Batf3*⁻/⁻ mice following repeat spore exposure (Supplementary Fig. 18B). Furthermore, pulmonary CD4⁺ T cell type 2 and type 17 cytokine responses were also unimpaired, or even elevated, relative to WT controls (Supplementary Fig. 18C), demonstrating that cDC1s are not necessary for induction of anti-*Af* allergic airway inflammation.

Together, these data show that Mgl2⁺ cDC2s play a crucial role in promoting type 2, but not type 17, fungal allergic airway inflammation, while cDC1s and pDCs are not essential for either type 2 or type 17 induction and development in this context.

## Discussion

Using complementary single cell technologies, along with deficiency or depletion of defined immune cell types in vivo, we have identified that a specific DC subset, namely Mgl2⁺ cDC2s, plays a critical role in promoting type 2, but not type 17, features of fungal allergic airway inflammation.

In keeping with previous reports that had suggested that CD4⁺ T cells are required for anti-*Af* allergic airway inflammation[10,12,71], we have clearly demonstrated that CD4⁺ T cells are vital for development of allergic airway inflammation against inhaled *Af* spores. Numerous studies have shown innate cell sources of type 2 and 17 cytokines (ILCs and γδ T cells) can be more impactful than CD4⁺ T cells in mediating allergic inflammation (including against fungi such as *Alternaria*)[5]. However, we found that it was CD4⁺ T cells, not ILCs, that were the overwhelming majority of type 2 cytokine secreting cells (Fig. 1). This was surprising, as *Af* proteases have been suggested to trigger ILC-induced eosinophilic inflammation independently of CD4⁺ T cells[33], and chitin (a vital component of the *Af* cell wall) has been shown to stimulate ILC-driven influx of IL-17 secreting γδ T cells into the lung[40]. Furthermore, it has been highlighted that γδ T cells can support type 2 pulmonary inflammation during helminth infection[72]. Our utilisation of cytokine reporter mice and CD4⁺ T cell depletion allowed us to determine that type 2 and type 17 cytokine expressing cells that underpinned fungal allergic airway inflammation, were critically dependent on CD4⁺ T cells (Fig. 2). Our results correlate well with studies on asthma cohorts which have highlighted that type 2 and type 17 fungal specific CD4⁺ T cells are prominent in severe fungal asthma patients[17]. Our use of γδ T cell deficient mice suggests that these cells do not have a dominant role in promoting anti-*Af* inflammation in the lung, and instead may actually regulate these responses. Although how γδ T cells achieve this remains unclear, previous studies have highlighted that they can express receptors (e.g. Dectin-1) that may directly sense fungi and can exhibit anti-*Af* activity[73,74]. Whether this activity explains how they limit allergic airway inflammation needs further study. Finally, some of the differences we have observed between *Af* triggered allergic airway responses and other allergenic fungi (e.g. ILC2 expansion in response to *Alternaria*[15]) could be due to low doses of

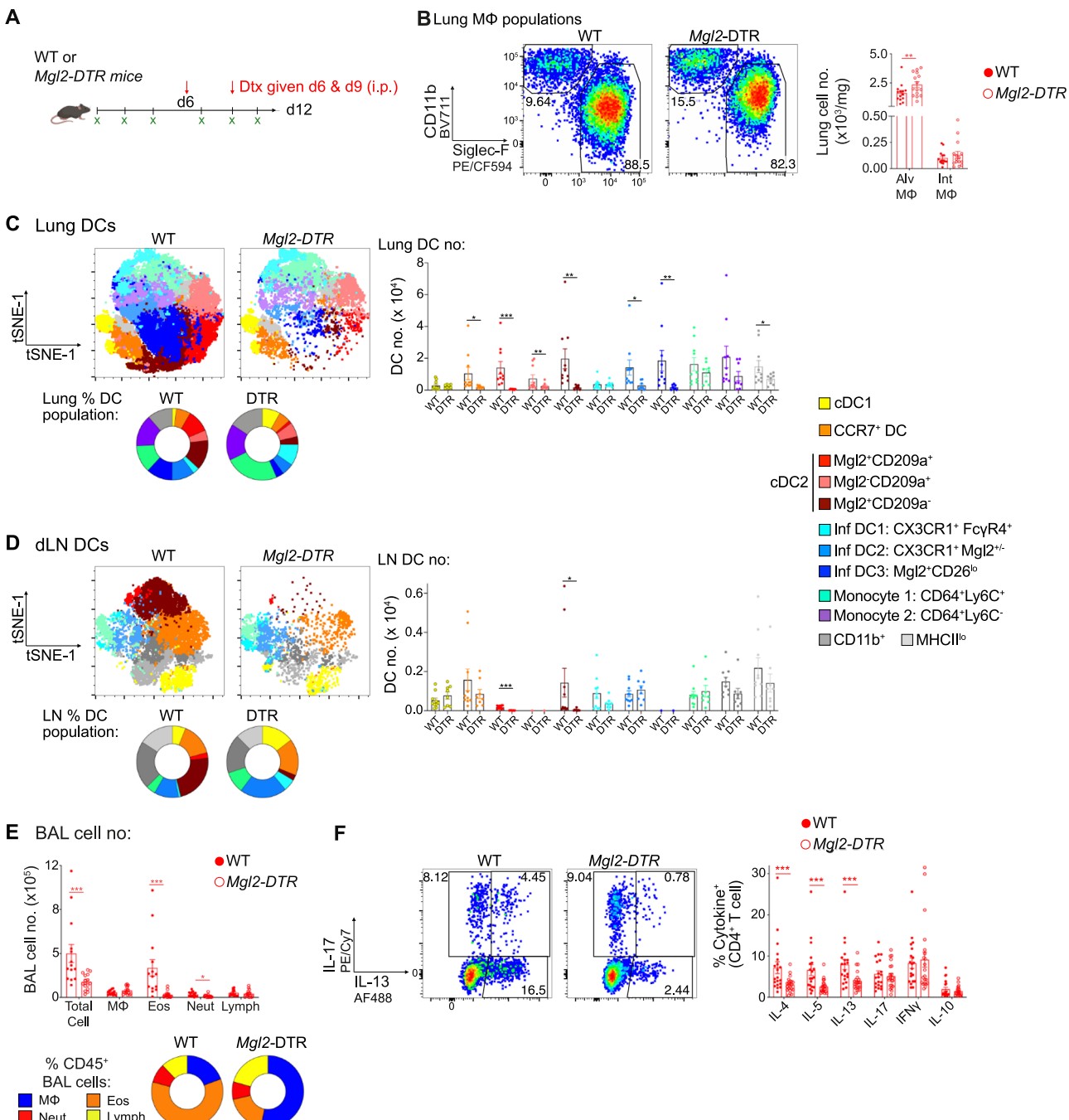

**Fig. 6 | Mgl2 cDC2s are crucial for promoting fungal type 2 allergic airway inflammation. A** WT or *Mgl2*-DTR mice were treated with diphtheria toxin (DTx) (on d6 and d9) whilst being exposed to repeat doses of *Af* spores (4 ×10⁵ per dose) BAL fluid and lung tissue were harvested 24 h after the final dose. Created in BioRender. Cook, P. (2024) https://BioRender.com/z24b189. **B** Representative flow cytometry plots identifying MΦ subsets and graphs show the number of MΦs in the lung tissue. **C, D** Representative flow cytometry tSNE plots show unbiased clustering analysis of identified subsets within the DC population that were detected in the dLN. Graphs show the number and percentage of DC subsets in the (**C**) lung and (**D**) dLN. **E** Graphs show the number and percentage of macrophages (MΦs), eosinophils (Eos), neutrophils (Neut) and lymphocytes (Lymph) detected in the BAL fluid 24 h after the sixth dose. **F** Representative flow cytometry plots identify IL-13⁺ and IL-17⁺ populations via intracellular cytokine staining of lung CD4⁺ T cells post stimulation with PMA/ionomycin. Graphs show the proportion of CD4⁺ T cells expressing type 1, type 2 and type 17 cytokine. **B, E** data from 3 independent experiments (*n* = 29 biologically independent animals). **C, D** data from 2

independent experiments (*n* = 18 biologically independent animals). **F** data from 4 independent experiments (*n* = 41 biologically independent animals). Data were fit to a linear mixed effect model, with experimental day as a random effect variable, and groups compared with a two-sided Tukey's multiple comparison test. *$p < 0.05$, **$p < 0.01$, ***$p < 0.001$. Data are presented as mean values ± SEM. **B** *P* values, Alv MΦ; WT vs *Mgl2-DTR* = 0.00403. **C** *P* values, CCR7⁺ DC; WT vs *Mgl2-DTR* = 0.0158. cDC2 Mgl2⁺CD209a⁺; WT vs *Mgl2-DTR* = 0.00021. cDC2 Mgl2⁻CD209a⁺; WT vs *Mgl2-DTR* = 0.00845. cDC2 Mgl2⁺CD209a⁻; WT vs *Mgl2-DTR* = 0.00311. Inf DC2; WT vs *Mgl2-DTR* = 0.027. Inf DC3; WT vs *Mgl2-DTR* = 0.00783. CD11b⁺; WT vs *Mgl2-DTR* = 0.0399. **D** *P* values, cDC2 Mgl2⁺CD209a⁺; WT vs *Mgl2-DTR* = 1.62 × 10⁻¹⁴. cDC2 Mgl2⁺CD209a⁻; WT vs *Mgl2-DTR* = 0.0332. **E** *P values*, Total Cells; WT vs *Mgl2-DTR* = 0.000503. Eos; WT vs *Mgl2-DTR* = 0.000149. Neut; WT vs *Mgl2-DTR* = 0.0131. **F** *P values*, IL-4; WT vs *Mgl2-DTR* = 0.000773. IL-5; WT vs *Mgl2-DTR* = 7.69 × 10⁻⁵. IL-13; WT vs *Mgl2-DTR* = 0.000636. Source data are provided as a Source Data File. Schematics in figures were created in https://BioRender.com.

*Aspergillus* spores being less disruptive to the epithelial barrier. For example, we could not detect alarmin cytokines such as IL-33 or TSLP in BAL fluid from *Af*-exposed mice (Supplementary Fig. 1E), while these key inflammatory mediators are produced in response to *Alternaria*[36]. Additional work is needed to directly address this hypothesis, and to determine whether γδ T cells and/or ILCs play supportive roles in CD4+ T cell mediated allergic airway inflammation against *Af* spores.

The central importance of CD4+ T cells in development of allergic airway inflammation in our experiments, along with our demonstration that depletion of CD11c+ APCs significantly reduced both type 2 and type 17 features of the pulmonary anti-*Af* response (Fig. 3), prompted direct assessment of the role of DCs, the most capable CD11c+ APCs for activation and polarisation of CD4+ T cells in the lung or elsewhere[5,21,57], in initiating the pulmonary response against *Af*. Our strategy of complementary use of single cell tools (scRNAseq, CyTOF and multi-parameter flow cytometry panels) allowed us to interrogate and delineate which lung CD11c+MHCII+ APCs were most dramatically influenced by *Af* spore exposure at different stages of allergic airway inflammation development. This approach revealed several clusters of DCs that have previously been noted in other inflammatory contexts[20] as being present during fungal allergic inflammation, identifying clusters of CCR7+ DCs, cDC2s and Inf DCs that all expanded in the lung during spore exposure (Figs. 4 and 5). Further, we found that cDC2s and CCR7+ DCs were the dominant clusters in the dLNs following repeat doses of *Af* spores (Fig. 5). Mass and flow cytometry showed that cDC2s could be separated based on expression of Mgl2 and CD209a, C-type lectin receptors associated with DCs that have putative type 2 and 17 polarising capabilities[42,43,68,75]. While Mgl2+ and CD209a+ cDC2s were found at similar frequencies in the lung, few CD209a+ cDC2s were detectable in the dLNs, where the majority were Mgl2+. Although the connection between these cDC2 clusters is unclear, splenic cDC2 subsets have previously been proposed to be linked to their expression of mRNA for RORγT and Tbet (called 'cDC2-A' and 'cDC2-B')[76]. However, neither RORγT nor Tbet were significantly expressed in any clusters in our scRNAseq dataset, in keeping with other scRNAseq studies that have similarly failed to identify RORγT or Tbet expression as signatures of lung cDC2 heterogeneity[44,47].

Informed by the above, targeted depletion of Mgl2+ cDC2s revealed the crucial role for these cells in supporting type 2 airway inflammation against *Af* (Fig. 6). The fact that Mgl2 depletion had little impact on Mgl2-CD209a+ cDC2, CCR7+ DC or Inf DC clusters, especially in the dLNs, suggests that they are not essential in this context. The reduction of Mgl2- DCs in lung tissue is likely an indirect effect of lower inflammation in Mgl2-depleted mice leading to reduced influx of these subsets. Although mRNA for Mgl2 was also expressed by some Inf DCs (Fig. 4), this was much less apparent at a protein level (Supplementary Fig. 15), and few Inf DCs were found in the dLNs during *Af* exposure cDC2s (Fig. 5). This is surprising, as previous work has identified that moDCs (which would now be referred to as Inf DCs) can regulate type 2 and type 17 responses in the lung when constantly challenged with high numbers of *Af* spores[24]. Furthermore, we did not observe any role for cDC1s (using *Batf3*−/− mice) or pDCs (using *Bdca2*DTR mice) in promoting either type 2 or type 17 features of fungal allergic airway inflammation (Supplementary Figs. 16 and 6), despite the fact that cDC1s and pDCs can respond to *Af* and support type 17 features of inflammation during invasive aspergillosis[22,23,30]. Indeed, we found that repeated *Af* exposure increased cDC1 expression of genes associated with metabolic and antigen processing pathways (Supplementary Fig. 12), showing that these cells are responding to *Af* in our experiments. Therefore, our data support a role for *Af*-activated cDC1s in inhibiting, rather than promoting, pulmonary Th2 responses after *Af* exposure (Supplementary Fig. 16C), in line with previous work suggesting cDC1 regulation of type 2 inflammation during invasive pulmonary aspergillosis and allergic models[29,77]. This highlights that the pulmonary DC populations we have identified via scRNAseq, CyTOF

and flow cytometry have distinct abilities to direct the character of airway inflammation against fungal allergens such as *Af*.

Another important question that arises from this study is what role the other (Mgl2-) DC clusters that expand following repeat *Af* exposure may play in fungal allergic airway inflammation, and whether one or more of these clusters are responsible for promoting pulmonary type 17 responses. Since both Mgl2+CD209a+ and Mgl2+CD209a- were efficiently depleted in *Mgl2*-DTR^het mice (Fig. 6), one candidate for this may be the Mgl2-CD209a+ cDC2s, as previous studies have shown that CD209a (the C-type lectin receptor SIGNR5, a homologue of human DC-SIGN) deficient mice display impaired type 17 inflammation during helminth infection[75]. In our experiments, Mgl2-CD209a+ cDC2s were not significantly affected by Mgl2 depletion, coincident with maintained type 17 inflammation. However, few CD209a+ cDC2s (whether Mgl2+ or Mgl2-) were evident in the dLNs at any stage following *Af* exposure (Fig. 6), indicating that they may not be critical for initial priming of type 17 pulmonary inflammation, though they may support these responses in the lung tissue. In addition to Mgl2-CD209a+ cDC2s, it is possible that CCR7+ DCs or Inf DCs could also be involved in promoting pulmonary type 17 inflammation against *Af*. However, while Inf DCs were impacted by Mgl2 cell depletion, CCR7+ DCs in dLNs were unaltered, making this population the most likely responsible for promoting type 17 inflammation against *Af*. Based on our scRNAseq data, CCR7+ DCs appeared to express mediators associated with Th17 responses, including *Ccl22* and *Ccl5*, recently proposed to be a major discriminator between high and low type 2 asthma patients[78]. Furthermore, a previous study identified a similar CCR7+ DC cluster and termed these as 'mregDCs' that can limit anti-tumour immunity[47]. Other reports have questioned whether CCR7+ DCs are a separate subset or have arisen from cDC1s or cDC2s[20]. Indeed, only by optimising our approaches for CCR7 staining (by using labelled CCL19 chemokine) were we able to clearly identify this population, which may have caused misidentification in previous studies relying on CCR7 staining.

Identification of Inf DCs (including the Mo cluster) through use of CD26 and CD64 expression agrees with previous work that proposed Inf DCs may be derived from cDC clusters and express Fcer1g[44], a suggestion supported by our scRNAseq data showing the highest expression of Fcer1g on the four Inf DC clusters[44] (though this was not confirmed at a protein level). Whether these Inf DCs have a role in promoting fungal allergic airway inflammation remains unclear, although the fact that they do not appear to be readily migrating to the dLNs suggests their role may be more to sustain immune events at the mucosal barrier site, perhaps via secretion of chemokines and cytokines (e.g IL-6, CCL6, CCL9, CCL24 and CXCL2), especially as they express several genes associated with MΦs in type 2 inflammation (e.g. *Retnla* and *Chil3*)[79]. Though we could identify three separate clusters of Inf DCs at both the gene and protein level, we cannot be certain that these are discrete subsets. For example, it could be possible that these different clusters are different maturation stages of one Inf DC subset[80]. New tools and strategies to selectively deplete these populations are required to address this issue.

Although our data show that Mgl2+ cDC2s are vital for anti-fungal type 2 inflammation, Mgl2+ cDCs have been implicated in promoting type 2 and type 17 inflammation in the skin (during helminth infection and psoriasis, respectively)[43,81]. A recent study has further highlighted that skin Mgl2+ cDC2s can induce Tregs against commensals in neonates[82]. Furthermore, Mgl2+ cDCs that drain from the lung into the dLNs have been implicated in promoting type 17 anti-bacterial immunity[68]. It remains unclear why Mgl2+ cDC2s are dispensable for pulmonary type 17 inflammation against *Af*. Additionally, while Mgl2+ cDCs are clearly crucial for fungal type 2 inflammation, the mechanism(s) by which they mediate such responses also remain unclear. Pathway analysis identified that *Af* exposure modified cDC2 phagosome and metabolic activity (Supplementary Fig. 12), both being

implicated in regulating DC microbe acquisition and processing[83], reminiscent of changes observed in MΦ mediated immunity against high dose *Af* in vitro and in vivo[84]. Further work is needed to address whether metabolic reprogramming is a critical regulator of Mgl2$^+$ cDC2 capability to trigger fungal allergic inflammation.

In summary, our work highlights the utility of combined mRNA and protein based single cell approaches to delineate DC subset heterogeneity during anti-fungal lung inflammation, and to inform depletion strategies to identify the relative importance of distinct DC subsets in orchestrating different facets of pulmonary allergic airway inflammation. In this way, our data provide further insight into the role of defined DC subsets in governing specific outcomes in pulmonary inflammation, that will be relevant for understanding key cellular and cytokine networks in fungal asthma and other respiratory diseases.

## Methods

### Experimental animals

*Batf3*$^{-/-}$[26], Bdca2$^{DTr}$[46], Cd11c.DOG[85], Mgl2-DTR[43] and *Tcrd*$^{-/-}$ mice were generated as previously described. Dual IL-13 and IL-17 reporter mice (*Il13*$^{eGFP}$*Il17*$^{Cre}$*ROSA*$^{eYFP}$)[35] were generated by crossing *Il13*$^{eGFP}$[35] with IL-17 *Il17*$^{Cre}$*ROSA*$^{eYFP}$[34]. All strains were on a C57BL/6 background and C57BL/6 mice were purchased from Envigo. For the majority of experiments using Mgl2-DTR het mice, these were generated with crossing Mgl2-DTR mice with Pep3 (congenic CD45.1 strain) mice. C57BL/6 x Pep3 mice were used as WT controls. Pep3 mice and C57BL/6 mice displayed comparable responses to repeat doses of spores. Mice were bred and maintained under specific pathogen-free conditions at The University of Manchester and The University of Exeter. Experiments in Manchester were approved under a project license granted (to A.S.M.) by the Home Office UK and by the University of Manchester Animal Welfare and Ethical Review Body (AWERB). Experiments in Exeter were approved under a project license granted (to P.C.C.) by the Home Office UK and by the University of Exeter AWERB. In both places, mice were housed in individually ventilated cages, with cage temperature at 23 °C, humidity at 54% and a 12 hr light/ dark cycle. All experiments were performed in accordance with the United Kingdom Animals (Scientific Procedures) Act of 1986.

### Mouse model of anti-fungal allergic inflammation

*Aspergillus fumigatus* spores (strain: CEA10[86]) were generated as previously described[87]. Briefly, spores were plated out on SAB agar. Ungerminated conidia were harvested and filtered, to remove debris and hyphal fragments. Conidia were washed extensively in sterile water before being resuspended in PBS containing 0.05% Tween-80. Aliquots of concentrated spore suspension were stored at −80°C. For each dose, a fresh aliquot was diluted so $4 \times 10^5$ (in 30–50 μl) spores in PBS 0.05% Tween-80 were given to mice via intranasal transfer. Mice were repeatedly exposed to spores or PBS 0.05% Tween-80 (3 doses a week) and tissues harvested 24 h after 3, 6 or 9 doses (exposure days 0, 2, 4, 7, 9, 11, 14, 16, 18 and harvests d5, d12 and d19). To measure fungal burden, lung lobes were collected and weighed. The lobes selected for CFU measurement were weighed and collected in 2 ml PBS, homogenised, suspensions spread on Potato Dextrose Agar (PDA) plates and incubated overnight at 37°C. *Af* colony forming units (CFU) from lung lobes were counted and CFU for the whole lung tissue calculated using weight information.

### Cell depletion strategies

CD4$^+$ T cells were depleted with a single dose of anti-CD4 mAb (clone GK1.5) or isotype control. Antibodies were administered to mice via an i.p. injection (250 μg per dose). Diphtheria toxin (DTx) was administered to CD11c.DOG, *Bdca2*$^{DTr}$ and *Mgl2*-DTR mice for cell depletion as previously described[41,43,88]. For CD11c.DOG mice, 8 ng/g DTx was administered daily for a six day period (started on day 6, one day prior to forth spore dose and continued to sixth spore dose, harvest 24 h

later). For *Bdca2*$^{DTr}$ mice 4-8 ng/g DTx was administered on d13, d15 and d17. For *Mgl2*-DTR$^{het}$ mice, 0.5 μg DTx was administered twice (on d6 and on d9).

### Enzyme-linked immunosorbent assays

ELISAs to detect Areg, IL-1β, CXCL1, CCL17, CCL22, GM-CSF, IL-33, TNF, TSLP, Ym1 (duosets from R&D systems), IL-4, IL-5, IL-6, IL-10, IL-12p40, IL-13, IL-17 IFNγ, (BioLegend), RELMα and Ym1/2 (PeproTech) were performed on BAL fluid as per manufacturers instructions. To detect anti-*Af* antibodies, *Af* antigen coated plates were generated by snap freezing spores in liquid nitrogen before lyophilisation for 5 d. Soluble antigens were prepared by mixing the lyophilised spores in PBS for 1 h (to a concentration of 1 mg/ml) and centrifuged for 10 min at $3220 \times g$. The collected supernatant was used to coat the wells (50 μL/well) of Maxisorp microtiter plates for 16 h at 4°C. The wells were then washed three times with PBST (PBS containing 0.05% (v/v) Tween-20), once with PBS (5 min), and given a final rinse with dH$_2$O before air-drying in sealed plastic bags. To detect *Af*-specific immunoglobulins, *Af*-antigen coated wells were incubated for 1 h with mouse serum double diluted in PBST (starting dilution of 1 in 50 (v/v)). Control wells were incubated with PBST only. The wells were then washed four times with PBST, and incubated for 1 h with either goat anti-mouse 13, IgM, IgA (H + L) horseradish peroxidase conjugate (PA1-84388, ThermoFisher) or goat anti-mouse IgE horseradish peroxidase conjugate (PA1-84764, ThermoFisher), both diluted 1 in 5000 in PBST. All incubation steps were performed at 23°C in sealed plastic bags. The plates were washed with PBST followed by a final wash of PBS. Bound antibody visualised by incubating wells with tetramethyl benzidine (TMB) substrate solution for 30 min, and stopped by the addition of 3 M H$_2$SO$_4$. Absorbance values were determined at 450 nm using a microplate reader (infinite F50, Tecan Austria GmbH). The threshold for detection of the antigen in ELISA was determined from the absorbance values of control well, which were consistently in the range of 0.050−0.100. Consequently, absorbance values ≥ 0.100 were considered as positive for the detection of antigen.

For determination of total IgE, mouse serum was double diluted in PBS in the wells of Maxisorp microtiter plates (50 μL/well), with a starting dilution of 1 in 50 (v/v). Immunoglobulins were immobilised by incubation at 16 h at 4°C, and the plates then washed and dried as described. The wells were incubated for 1 h with goat anti-mouse IgE horseradish peroxidase conjugate (PA1-84764, Invitrogen), washed with PBST followed by PBS, incubated with TMB substrate for 30 min. and stopped by the addition of 3 M H$_2$SO$_4$. Absorbance values were determined at 450 nm.

### RNA extraction and RT-qPCR

Lung tissue was immediately placed in RNAlater (ThermoFisher) and stored at −80°C. Tissues were homogenised with a Geno/Grinder (SPEX) and RNA isolated using RNeasy columns (Qiagen) following manufacture instructions. Complementary DNA was generated from extracted RNA using SuperScript-III Reverse Transcriptase and Oligo-dT (ThermoFisher) following manufacture instructions. Relative quantification of genes of interest was performed by qPCR analysis using QuantStudio Pro 7 system (ThermoFisher) and PowerTrack SYBR Green master mix (ThermoFisher), compared with a serially diluted standard of pooled cDNA. Expression was normalised to *Hprt* (primers in Supplementary Table 1).

### Histology

Lungs were collected in 10% neutral buffered formalin (NBF) (Sigma), incubated overnight at room temperature, with NBF removed and samples stored in 70% ethanol the next day. Lungs were dehydrated overnight with a Leica ASP 300 Tissue Processor (Leica Biosystems), embedded in paraffin and sectioned 5 μm thick. H&E staining was performed utilising a Leica autostainer (Leica). For AB-PAS (Alcian blue

−periodic acid Schiff) staining slides were rehydrated using a Leica autostainer (Leica), stained with Alcian blue (Sigma) for 15 minutes, 1% periodic acid (Sigma) for 5 minutes, Schiff's reagent (Sigma) for 10 minutes, Papan haematoxylin (Merck) and 5% acetic acid (Fisher) for 30 seconds. For Masson's trichrome staining slides were rehydrated using a Leica autostainer (Leica), stained with saturated aquarous picric acid (Sigma) for one hour, then stained with Weigart's iron hematoxylin (Sigma) for 10 minutes, then Biebrich scarlet-acid fuchsin (Sigma) for 10 minutes, then phosphotungstic/phosphmolybdic acid (Sigma) for 10 minutes, then 1% acetic acid (Sigma) for 2 minutes. After each step slides were washed with distilled water. Following staining all slides were dehydrated and mounted on the Leica autostainer and Leica CV5030 coverslipper (Leica). A Panoramic250 slide scanner (3D Histec) was used to obtain images. Image analysis was performed utilising ImageJ (NIH). For epithelial thickness measurements with H&E stained lung histology samples a total of 100 measurements were taken from airways of defined sizes (100–300 µM). For goblet cell hyperplasia measurements with AB-PAS stained samples airways 100–300 µm were chosen and mucin secreting goblet cells (purple) counted. Goblet cell counts were normalised to the airway circumference. To quantify collagen deposition on chosen airways (100–300 µM) in Masson's a Hue Saturation Brightness threshold on ImageJ was chosen to select only blue pixels, representing stained collagen. These pixels were quantified and expressed as a percentage of total lung pixels.

### Isolation of immune cells from the BAL, lung and dLN

Following killing, BAL cells were obtained by washing the lung airway with PBS containing 2% FBS and 2 mM EDTA (Sigma). Lungs were processed as previously described[54], briefly lung tissue was chopped and incubated at 37°C whilst shaking for 30 min with 0.8 U/ml Liberase TL and 80 U/ml DNase in HBSS (all Sigma). In contrast, dLN (not disrupted prior to incubation) were incubated at 37°C whilst shaking for 30 min with 1 U/ml Liberase TL and 80 U/ml DNase in HBSS (all Sigma). To stop digestions, ice cold PBS containing 2% FBS and 2 mM EDTA was added and then passed through a 70 µM cell strainer.

### Lung cell stimulation, flow cytometry staining and acquisition

For measurement of immune cell cytokine secretion, lung single cell suspensions were incubated with PMA (30 ng/ml, Sigma), Ionomycin (1 µg/ml, Sigma) and GolgiStop (BD) for 3 hours prior to surface and then intracellular staining. For surface staining, equal numbers of cells were stained for each sample, washed with ice-cold PBS and stained with Live/Dead Blue (ThermoFisher) or Zombie UV dye (BioLegend) for 10 min at room temperature. Samples were then incubated with αCD16/CD32 (2.4G2; BD Biosciences) in FACS buffer (PBS containing 2% FBS and 2 mM EDTA) before staining for surface markers at 4°C for 60 min (antibodies and labelled chemokines listed in Supplementary Data 1). Post staining, cells were washed twice in FACS buffer and then fixed in 1% paraformaldehyde in PBS for 10 min at room temperature. For detection of intracellular antigens, cells were processed with Foxp3 buffer staining set (ThermoFisher) and stained with selected antibodies. Samples were acquired on a BD Fortessa or BD Symphony with FacsDiva (BD) software. For most experiments, we counted total cells recovered from BAL fluid or lung tissue digestions using a hemocytometer and trypan blue, to exclude dead cells, prior to staining for flow cytometry. In some experiments CountBright Absolute Counting Beads (ThermoFisher) were added to samples prior to flow cytometry acquisition to calculate absolute cell counts.

### CyTOF antibody conjugations and staining

Pre-conjugated metal labelled antibodies were purchased from Standard Biotools, while custom conjugations were performed within the University of Manchester's Flow Cytometry Core Facility using the MAXPAR® antibody labelling kits from Standard Biotools according to

the manufacturer's instructions. Briefly, 5 µl of 50 mM stock lanthanide metal solutions were incubated with the MAXPAR® X8 polymer in L buffer for 60 mi at room temperature, before being washed with C buffer by centrifugation through 3 kDa Amicon® spin filters (Merck Millipore). In parallel, 100 µg of purified, carrier free antibody was reduced for 25 minutes at 37°C in R buffer containing 4 mM tris(2-carboxyethyl)phosphine (TCEP) solution (Thermo) diluted in R buffer. The reduced antibody was washed in C buffer in 50 kDa Amicon® spin filters (Merck Millipore) before being mixed with the preloaded X8 polymer and incubated for 60–90 min at 37°C. Antibody/polymer complexes were washed four times in W buffer before assessment of antibody recovery was performed using a Nanodrop at Abs 280. Labelled antibodies were stored in Candour Stabilisation buffer (Candour Biosciences GmbH) at -0.5 µg/mL at 4°C. Antibodies labelled with isotopically pure cisplatin (194Pt and 198Pt, Standard Biotools) were processed as described[89] where 20 µl of 1 mM stocks were added to reduced antibodies in a final volume of 1 mL of C buffer, incubated, washed and stored as described above. Platinum-conjugated antibodies were generated as previously described[89].

Samples were stained following the flow cytometry staining protocol. Cells were washed in PBS before incubation with 1.25 nM Cisplatin solution prior to addition of αCD16/CD32 (2.4G2; BD) in FACS buffer (PBS containing 2% FBS and 2 mM EDTA) before staining for two rounds of surface markers at 4°C for 60 minutes (antibodies and labelled chemokines listed in Supplementary Table 2). Post staining, cells were washed twice in FACS buffer and then fixed in 1% paraformaldehyde in PBS for 10 min at room temperature. For detection of intracellular antigens, cells were fixed with Foxp3 buffer staining set (ThermoFisher) and stained with selected antibodies. Once stained samples were frozen in 10% DMSO (Sigma) 50% FBS and RPMI prior to being run on the CyTOF Helios. Frozen samples were thawed on ice and washed once in MAXPAR PBS, and twice in milliQ water before being resuspended in MilliQ water at 0.5 x 10^6 cells / mL. EQ beads for post-acquisition normalisation were added to the sample and data was acquired on a Helios mass cytometer (Standard Biotools). Resultant *.FCS files were normalised with the CyTOFv2 software before downstream analysis.

### Flow cytometry and CyTOF DC clustering analysis

Samples acquired via flow cytometry were analysed with FlowJo v10 (Tree Star). Flow cytometry gating schemes that were utilised to identify cell populations are shown in Supplementary Figs. (1, 2, 5, 6, 13, 17). For all flow cytometry analysis, dead cells and doublets were removed as highlighted in Supplementary Fig. 13B.

**DC clustering analysis.** on CyTOF and flow cytometry and was performed independently using FlowJo v10 (Tree Star). The DC population was clearly identified through sequential gating with both CyTOF and flow cytometry data (Supplementary Fig. 13). The DC population of each sample within each experiment were concatenated to one FCS file. tSNE and FlowSOM clustering analysis[64] was then performed on this concatenated FCS file. Markers only associated with DC populations were included for the clustering analysis (Figs. 5 and 6, Supplementary Figs. 14, 15 and 16). Clusters were assigned to different populations based on marker expression (Supplementary Figs. 14, 14 and 16). After identification, individual samples were extracted from the concatenated file to identify the proportion and number of DCs within each cluster across each sample.

### Single-cell isolation, library construction and sequencing

Lung samples from naïve or *Af*-exposed mice were digested as mentioned above and OptiPrep (Stemcell Technologies) gradient was used to enrich DCs. Briefly, lung single cell suspension was resuspended in 15% OptiPrep then 11.5% OptiPrep followed by HBSS was layered on top to form distinct layers. Gradients were spun at 600 g for 15 min with no

brake and a band of mononuclear cells at the top of the gradient was collected. DCs were then isolated via flow sorting (CD3⁻CD19⁻CD45R⁻Ly6G⁻MerTK⁻NK1.1⁻Siglec-F⁻Ter119⁻, CD45⁺CD11c⁺IA/IE⁺) using an Influx (BD BioSciences) to a purity of 95-99%. Gene expression libraries were prepared from single cells using the Chromium Controller and Single Cell 3' Reagent Kits v2 (10x Genomics) according to the manufacturer's protocol, cDNA libraries were sequenced on the Illumina HiSeq4000.

## Single cell pipeline and analysis

**Data pre-processing.** FASTQ sequence files generated from the sequencer were processed using 10x Genomics custom pipeline Cell Ranger v2.0.2. This pipeline generated the fastq files which are then aligned to the mm10 custom genome with all the default parameters of cellranger. The pipeline then identified the barcodes associated with cells and counted UMIs mapped to each cell. Cellranger uses STAR aligner to align reads to genome so it discards all the counts mapping to multiple loci during counting. The uniquely mapped UMI counts are reported in a gene by cell count matrix represented as sparse matrix format. We used the cellranger's aggr command to aggregate the samples from the three samples. We used the default down-sampling parameter of cellranger aggr that down sample reads to make the average read depths per cell equal between the groups. We identified 2539 cells across all three samples before quality filtering.

**Filtering.** We removed the low-quality cells from the dataset to ensure that the technical noise do not affect the downstream analysis. We looked into three commonly used parameter for cell quality evaluation, the number of UMIs per cell barcode (library size), the number of genes per cell barcode and the proportion of UMIs that are mapped to mitochondrial genes. Cells that have lower UMI counts than three Median Absolute Deviation (MAD) for the first two metrices and cells having higher proportion of reads mapped to mitochondrial genes with a cutoff of five MADs are filtered out. We then plotted violin plots for these three metrices to see whether there are cells that have outlier distributions which can indicate doublets or multiplets of cells. We removed the outlier cells that have total read counts more than 30,000 as potential doublets. After these filtering, 2357 cells (554 sample Naïve, 687 sample *Af* d5, 1116 *Af* 12) cells remained for downstream analysis.

**Classification of cell-cycle phase.** We used the cyclone method[90] to calculate for each cell the score of G1 phase and G2M phase. Cyclone uses a pre-build model that is trained on a known cell-cycle dataset where the sign of difference in expression between a pair of genes was used to mark the cell-cycle stages. The cell cycle phase of each cell is identified based on the observed sign for each marker pair of each of the phases.

**Gene filtering and Normalisation.** We then filtered out the genes with average UMI counts below 0.05 as we assume these low-abundance genes do not give much information and are unreliable for downstream statistical analysis[91]. After this filtering 8848 genes were left for downstream analysis. In order to account for the various sequencing depth of each cell, we normalized the raw counts using the deconvolution-based method[90]. In this method, counts from many cells are pooled together to circumvent the issue of higher number of zeros that are common in single cell RNA-seq data. This pool-based size-factors are then deconvoluted to find the size factor of each cell. These normalized data are then log-transformed with a pseudo-count of 1.

## Visualisation and clustering

The first step for visualisation and clustering is to identify the Highly Variable Genes (HVGs). To do this, we first decomposed the variance of each gene expression values into technical and biological components and identified the genes for which biological components were significantly greater than zero. We call these genes as HVG. We identified 124 HVG genes in our dataset. These HVG genes were then used to reduce the dimensions of the dataset using PCA. The dimension of dataset were further reduced to 2D using t-SNE and UMAP, where 1 to 14 components of the PCA were given as input. The cells were grouped into their putative clusters using the dynamic tree cut method[92]. Dynamic tree cut method identifies the branch cutting point of a dendrogram dynamically and combines the advantage of both hierarchical and K-medoid clustering approach. This method identified 11 clusters in the aggregated population.

## Identification of marker genes

To identify the marker gene for a cluster, we compared that cluster with all other clusters. We then report the genes that are differentially expressed in that cluster as the marker for the cluster. We used the edgeR tool to identify these marker genes[93]. These marker genes were then used to annotate the cell types of a cluster. As the dataset had multiple samples and cluster, we formulated different combinations of samples and clusters and identified marker genes for these combinations.

## Gene pathway analysis

Gene-set enrichment analysis was performed using the 'clusterProfiler' package for R. Differential expression of genes between experimental time points was calculated by Welch's t-test and adjusted for multiple comparisons by calculating the false discovery rate. For each DC cluster and experimental comparison the $\log_2$(fold change) of each gene was calculated and ordered by value following which enrichment analysis was performed using the 'compareCluster' function with fun = "gseKEGG" and pvaluecutoff = "0.05". Pathways were subsequently filtered so that a pathway was considered significantly differentially enriched when p.adjust < 0.01 & absolute enrichment score > 0.5. Nonsensical pathways were manually filtered out prior to ranking. Significant pathways were ranked by maximum absolute enrichment score across all DC clusters and experimental comparisons.

## Statistics

Data are shown as mean values ± SEM. To detect significant differences, mixed linear models were fitted utilising Rstudio (R Core Team)[42]. To account for random variation, the experimental day was designated a random effect variable, with timepoint or genotype as fixed effect variables. To compare multiple groups a post-hoc Tukeys HSD test was used.

## Reporting summary

Further information on research design is available in the Nature Portfolio Reporting Summary linked to this article.

## Data availability

The scRNA-seq data generated in this study have been deposited on the ArrayExpress database under accession code E-MTAB-13740. All other data are available in the article and its Supplementary files or from the corresponding author upon request. Source data are provided with this paper.

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

## Acknowledgements

We thank members of the Lydia Becker Institute and MacDonald laboratory (University of Manchester) for scientific discussions and some experimental assistance, and the University of Manchester flow cytometry facilities. We would also like to thank Prof Elaine Bignell for supplying *Aspergillus* strains and helpful discussions when establishing the study, as well as Prof Ken Murphy and Prof Akiko Iwasaki for sharing *Batf3*$^{-/-}$ and *Mgl2*-DTR mice, respectively. PCC was supported by funding from University of Manchester Dean's Prise Early Career Research Fellowship, Springboard Award (Academy of Medical Sciences, SBF002/1076) a Wellcome Trust Sir Henry Dale Fellowship (218550/Z/19/Z), the MRC Centre for Medical Mycology at the University of Exeter (MR/N006364/2 and MR/V033417/1), and the NIHR Exeter Biomedical Research Centre (NIHR203320). ELH was supported by a BBSRC CASE studentship (with GSK) (BB/P504543/1). DC was supported by a MRC Doctoral Training Grant MR/P501955/2. JEA was supported by MRC-UK (MR/V011235/1) and the Wellcome Trust (106898/A/15/Z). ASM was supported by funding from GSK, the Lydia Becker Institute and the MRC (MR/W018748/1). The views expressed are those of the author(s) and not necessarily those of the NIHR or the Department of Health and Social Care. Schematics in figures were created in https://BioRender.com. For the purpose of open access, the author has applied a 'Creative Commons Attribution (CC BY) licence to any Author Accepted Manuscript version arising from this submission'.

## Author contributions

P.C.C. and A.S.M. were responsible for conceptualisation. P.C.C., S.L.B., E.L.H., J.F-S., D.C., S.C., S.B., F.R.S., G.H., C.R.T., conducted or enabled investigations M.B., L.B., J.E.K., C.R.T., J.E.A. provided resources. P.C.C. wrote the manuscript with input from all other authors. P.C.C. and A.S.M. were involved in funding acquisition.

## Competing interests

Individuals based at the Lydia Becker Institute received funding from GSK. These authors (P.C.C., S.L.B, E.L.H., F.R.S., A.S.M.) declare that the research was conducted in the absence of any commercial or financial relationships that could be construed as a potential conflict of interest. The remaining authors declare no competing interests.
