## [Transparent Peer Review file · Nature Communications]

Mgl2+ cDC2s coordinate fungal allergic airway type 2, but not type 17, inflammation

Corresponding Author: Dr Peter Cook

Version 0:

Reviewer comments:

Reviewer #1

(Remarks to the Author)

The authors present a detailed analysis of dendritic cell subsets accumulating in the mouse lung and associated lymph nodes after challenge with low-dose *Aspergillus fumigatus* spores at varying time points. They use single cell RNA sequencing, multiparameter flow cytometry and flow mass cytometry to confirm the accumulation, over time, of the same, multiple DC subsets in the lung, but differential accumulation of DCs in regional lymph nodes, most especially Mgl2+cDC2. They further show functionally using depletion strategies that Mgl2+cDC2 are primarily responsible for Th2, but not Th17, development in the context of fungal spore challenge. The studies revisit previously well-tread ground demonstrating, for example, that fungus-induced allergic airway disease in mice primarily consists of type 2 and type 17 inflammation and that such inflammation is due primarily to CD4 T cell-derived cytokines, with minimal or no contribution from other cells such as T cells, ILC, and others. Although the information is redundant, it is important to keep as the findings counter so many more recent reports that portray ILC as essential effector cells in similar models and that ILC are the predominant sources of type 2 and type 17 cytokines in various allergic contexts. The first few figures therefore restore the primacy of CD4 T cells as primary drivers of the allergic airway disease phenotype. The report consistently uses only the strongest methodology. Major findings are highly original and very important to the field scientifically and moreover portend therapeutic significance.

Specific comments and concerns:

1. The Results section carries forward and repeats introductory material; please reserve this section just for results, by beginning on line 111
2. Figure 2: the second graph from Fig.2A and 2C shows a bar under the first 4 and 3 items identified on the X axis terms and underneath the bar is the word "T cell". It is unclear what this is trying to designate, and no explanation is offered in the figure legend. Please remove this confusing part of the graphs or explain what it means if it is felt to be important.
3. The violin plots indicated in the Figure 4D legend are not shown in the figure-please correct. Presumably these were moved to Suppl. Fig. 7?
4. Figure 4/color schemes: This complex figure requires careful rewording to make clear what is going on. When referring to colors, please explicitly indicate what is being referred to: the dots (i.e., individual cells) or the ellipses surrounding groups of dots (presumably distinct DC phenotypes). Similarly, for most figures, the legends explaining the various heat maps is missing, especially for the supplementary figures. Thus for each use of heat maps for both RNA sequencing and cytof, please include the legends that explain the level of gene or protein expression.
5. Please apply tSNE and violin plot analyses to Suppl. Figure 10 if possible.
6. Gene nomenclature. Clec10a and Mgl2 are shown as distinct genes in Figure4 and discussed as such elsewhere. However, depending on the source, Clec10a (CD301) is usually reserved for the human ortholog, but the mouse version exists in two forms, Mgl1 and Mgl2 (CD301a and CD301b, respectively). Thus, please clarify what genes are actually being assessed in Fig. 4 and elsewhere when Clec10a/MGL2 are referred to. Please consider using systematic (i.e., CD) nomenclature consistently throughout for all genes to avoid such confusion.
7. The authors emphasize the antifungal nature of the allergic inflammation ("anti-fungal" is used 27 times in the manuscript), and yet no antifungal activity is shown. To help justify this, please consider adding physiologically relevant experiments testing whether depletion of Mgl2+cDC2 impairs fungal clearance from mouse lungs.

Reviewer #2

(Remarks to the Author)

In this study, the authors investigated immune response during fungal infection of the lung. It is well documented that *Aspergillus* infection triggers both Th2 and Th17 immune responses in the lung. The authors examined the role of specific dendritic cell (DC) subsets in induction of a Th2 vs a Th17 immune response. The study is well done and use of scRNA-seq in combination with genetically altered mice shows a role for a specific DC subset, Mgl2+ (CD301+) cDC2s, in promoting Type 2 but not Type 17 immune response. The key finding associating this DC subset with Type 2/Th2 inflammation, however, is not entirely novel and has been reported in previous studies, as the authors have also noted (refs 41, 50 and 76).

Major comments:

It is interesting that in their model, ILC2s were not found to be particularly important in inducing eosinophilic/Type 2 inflammation as opposed to many studies showing their involvement during infection by another fungus, *Alternaria*. This aspect needs more discussion. Do the two types of fungus have different effects on airway epithelial barrier disruption because of differential protease or other effector functions?

A previous paper (ref. 42) showed that moDCs, distinguished by Ly6C expression (which they propose are macrophages rather than DCs), do not migrate to lung-draining LNs, in contrast to what was previously published by the same authors. These authors also found that Mgl2+ DCs are only a minor population of all cDC2s in their models, which did not include *Aspergillus* infection. It will be helpful if the authors discussed the function of Mgl2+/CD301+ cDC2s in relation to other types of cDC2s in the context of their own study and the current literature.

The Inf-cDC2s with a hybrid phenotype with some features of cDC1s (ref. 42) was found to be type I IFN-dependent. Can the authors detect such a signature in any of the DC subsets in their study since these cDC2s were detected in multiple models of infection?

Lines 489-492-the message is not clear as written. The data show 3 types of cDC2s one of which co-expresses Mgl2 and CD209a. What is meant by connection between Mgl2+ and CD209a+ DCs? It will be helpful to run a Pseudotime analysis to determine relationships between the cell populations.

Figs. 3B and 3D-DT treatment abolished Mgl2+ DCs. However, the data show only a 50% loss of IL-13+ CD4+ T cells. These results are inconsistent with Mgl2+ cDC2s being mainly responsible for driving a Th2 response.

Line 513-role of DC subset that promotes a Type 17 immune response to *Aspergillus* infection. Ly6c+CD11b+TNF α + DCs were previously implicated in promoting a Th17 response and neutrophilic inflammation during *Aspergillus* infection (PMID: 21402950). Do the Mgl2+CD209a- DCs express TNF α ? The authors should examine expression of this cytokine in the DC subsets in the scRNA-seq data. This will be especially interesting since TNF- α , as in this study and others, has been associated with Th17 and neutrophilic inflammation. Did they detect an increase in TNF α and IL-17 in the CD11c.DOG mice where lung eosinophil numbers were reduced but neutrophil numbers increased?

In the Introduction and Results sections, the authors have made statements like the immune response to fungal infection is poorly understood (lines 36-38, 63-64; 109-111). The authors should cite previous well-cited papers on the contributions of specific types of DCs in inciting Th17-driven inflammation during *Aspergillus* infection (PMIDs: 21402950; 26365185). It will help the readers to expand their discussion their findings in light of these previous studies. Of note, some of their data on the contribution of CD4+ T cells vs T cells were previously studied by convention flow cytometry with essentially similar results.

Were mice of both sexes used in the study?

Minor issues:

Line 279, 280-unlike monocytes, macrophages are not circulating cells.

Line 490- the article "a" needs to be deleted.

The colors of CCR7+ DCs and Mgl2-CD209a+ DCs are difficult to distinguish in Fig. 5 and associated supplementary Figs.

Line 509-the authors should also cite the study showing a role for CD103+ DCs in promotion of antigen-induced tolerance (PMID: 23733880).

Reviewer #3

(Remarks to the Author)

In this study, the authors aim to identify the main cell types involved in the development of fungal allergic airway inflammation. Using a mouse model for fungal allergic airway inflammation, transgenic mouse models, and complementary single-cell technologies, they demonstrate that Mgl2+ cDC2s are critically involved in the development of type 2, but not type 17, fungal allergic airway inflammation.

Overall, this study presents a thorough characterization of key immune cell subsets implicated in fungal allergic airway

inflammation. Although fungal spores are widely spread in our environment and can trigger asthmatic initiation, the underlying mechanisms are largely elusive.

However, the following points need to be addressed to improve the comprehensibility of the study:

Results:

1. In this study, the mouse model for *Aspergillus fumigatus* (Af) spore-induced fungal airway inflammation has been extensively used and applied on both wild type and transgenic mice. Although a thorough description of the elicited immune cell responses is provided, no information on other phenotypic features of the disease is given. To highlight the relevance of the findings, potential phenotypic alterations, including airway hyperresponsiveness, peribronchial inflammation, and tissue remodelling, in the Af-exposed wild type and transgenic mice should be mentioned.
2. To perform a detailed characterization of the immune cell subtypes involved in fungal allergic airway inflammation, several transgenic mouse models were used. Were the included wild type control mice respective littermate controls, as appropriate? This information should be included.
3. Using Tcrd^{-/-} mice and the mouse model for *Aspergillus fumigatus* (Af) spore-induced fungal airway inflammation, the authors conclude that IL-17 secreting $\gamma\delta$ T cells are not essential for the development of pulmonary inflammatory responses against Af. However, based on Fig. 2C and 2D, $\gamma\delta$ T cell deficiency resulted in higher inflammatory responses upon Af exposure. In this Reviewer's opinion, this result is only shortly mentioned and not sufficiently discussed by the authors.
4. To study the role of CD11c⁺ cells in fungal airway inflammation, CD11c.DOG mice were used. As mentioned, diphtheria toxin administration resulted in depletion of CD11c⁺ alveolar macrophages, plasmacytoid dendritic cells (DCs), conventional DC1s (cDC1s), and Mgl2⁺ cDC2s. However, lung tissue MHCII+CD11c+CD11b+ cDC2s were not depleted, as mentioned in lines 200-202. How can this unexpected finding be explained?
5. Line 247-248: If understood correctly, this experiment was performed in wild type mice. This information should be provided clearly in the text.
6. Although Mgl2 is expressed in lung macrophages and especially in interstitial ones, as already mentioned in lines 410-411, diphtheria toxin administration to Af-exposed Mgl2-DTR mice did not affect the number of alveolar and interstitial macrophages compared to Af-exposed wild type mice, while Mgl2⁺ DC subsets were reduced. How can this be explained? To better understand such unexpected findings and the nature of these transgenic mouse models (see also comment 4), a characterization of the immune cell alterations upon diphtheria toxin administration alone and PBS (instead of Af) exposure would be useful.

Methods:

7. A more detailed description of the mouse model of anti-fungal allergic airway inflammation, including the timepoints of Af exposure, the use of control mice etc., should be provided in the "Methods" section.
8. In lines 632-633, the authors mention that CountBright Absolute Counting Beads were used in some experiments. Since absolute cell counts are provided in all experiments and respective figures, how were cell numbers counted in all cases if Beads were only applied in some experiments?

Figures /Figure legends:

9. Although sometimes not clearly stated in the "Results" section, it seems that immune cell characterization was not always performed at the same timepoint of Af exposure, based on the figure legends. For example, the analysis depicted in Supplementary Figure 6 regarding the comparison of wild type and Bdc2DTR mice was performed after nine Af doses. However, the comparison of wild type and Cd11c.DOG mice depicted in Supplementary Figure 5 was performed after six Af doses. Why were different timepoints chosen?
10. Minor language mistakes can be found in figure legends, e.g., in the description of Fig.2A, the word "prior" should be deleted. Additionally, all abbreviations should be explained appropriately (e.g., macrophages (M Φ), etc.). Careful proofreading of the text and figure legends of both main and supplementary figures is required.

Reviewer #4

(Remarks to the Author)

In the paper by Peter Cook et al, the authors have investigated the immunological pathways that underlie the mouse *Aspergillus fumigatus* conidia model of allergic airway inflammation. The authors used several genetically altered mouse lines to demonstrate that the underlying type 2 and type 17 inflammation in this model was dependent on the presence of both CD4⁺ T cells and CD11c⁺ cells, and more specifically, the Mgl2⁺ cDC2 subpopulation was essential for the type 2 response. In vivo immunological analysis was complemented by single cell RNA sequencing and additional CyTOF analysis. Overall, the data is convincing and the authors clearly show the dependence of allergic airway inflammation on the aforementioned immune populations in vivo.

The paper relies solely on immunological analysis, and no data is provided on the pathophysiology; including, but not limited to, airway hyperreactivity, serology and lung histology. Inclusion of this data would future strengthen the manuscript and put the immunological data into physiological context.

Along these lines, the authors need to show that the level of model induction is the same prior to intervention strategies e.g. before anti-CD4 or Dtx treatment. This can be done by including circulating serology data at the intervention and analysis time points.

Mgl2⁺ cDC2s have already been shown to initiate type 2 immunity, therefore, it not so surprising that they also act similarly in this model. One open question remains, which is the DC population which drives the type 17 response in this model? Can the authors shed some light on this population using the current data and/or in vitro co-culture experiments (e.g. isolated DC populations and T-cell response assays)?

Unless I am mistaken the conclusion that Mgl2⁺ cDC2 do not coordinate type 17 inflammation, is based on one graph in Figure 6E, namely the lack of reduced IL-17 levels following removal Mgl2⁺ cDC2. Can the authors provide independent

confirmation of this observation?

The single RNA analysis is a welcome addition to the paper, however, the authors have failed to use it to its full potential, and the manuscript can be improved by deeper analysis.

Please include gene enrichment and pathway analysis, which would give the reader additional insights into the behaviour of the different DC populations.

Do the different DC populations e.g. Inf DC1, DC2 etc. represent different lineages or cell states?

It is unclear how many individual mice were included in this analysis? Please specify. How were AlVmp excluded /alternatively, not identified in the scRNA analysis? Presumably, the entire scRNA dataset was concatenated prior to analysis, how did the cell/population abundance change in the different clusters over the different groups? Here, Fig 4C should be split by origin and shown as individual panels.

Legends need to be added to the tSNE plots (e.g. Supplementary Figures 7-9).

In Supplementary Figure 3D, Fig 2D, Fig 3D etc, only representative plots for IL-17 and IL-13 are shown, please include representative gating plots the remaining cytokines; IL-4, IL-5, IL-17, IFN γ and IL-10

Please include in the main manuscript Figure 6, the lung DC no. Please discuss why removing Mgl2, changes the abundance of multiple lung DC populations and not just those expressing Mgl2.

Minor points

Gating strategies should also include the initial gating steps e.g. FSC/SSC, singlets and viability plots.

Please include accession numbers for the qPCR primers in Supplementary Table 1, and additionally include the sequences for Hprt.

Please remove all axis breaks in the y-axis e.g. Fig 2C etc. The authors could consider using data transformation (e.g. log) to enable all data points to be visible.

Please clarify in the manuscript what is meant by "Individual samples were then extracted from concatenated files, ensuring that tSNE and FlowSOM clustering analysis was consistent across samples, to identify the proportion of DCs in each sample."

What exactly was done here and how did it ensure consistent clustering?

Version 1:

Reviewer comments:

Reviewer #1

(Remarks to the Author)

The authors have satisfactorily addressed all concerns; no further concerns are noted.

Reviewer #2

(Remarks to the Author)

The authors have responded in detail to the comments of all the reviewers. The strength of the study is the detailed characterization, using complementary approaches, of various DC subsets in the lungs and dLNs of mice subjected to the model of repetitive infection with Af spores. They show the selective migration of Mgl2+CD109- cDC2s and CCR7+ DCs to the dLNs which suggests their role in T2 priming. As also mentioned in my previous comments (and the authors agree) that the role of Mgl2+cDC2s in Th2 development has been shown previously although they may not have been in response to Af infection per se. There is no additional insight into what mechanism in a cell-intrinsic and selective fashion equips these DCs to promote Th2 development. Is it more their migration or they actually have Th2-promoting pathways activated in them? In other words, by experimental manipulation, if one forces the other similar subset, Mgl2+CD109+ cDC2s, to migrate to the dLNs, would they also be able to induce Th2 development? They discuss briefly upregulated phagosome activity but that is anticipated if the DC has to take up the fungal spores and process the Ags and they acknowledge such activity being observed in macs exposed to Af. Overall, while the study is well executed the information gained is somewhat incremental.

Reviewer #3

(Remarks to the Author)

I have reviewed the revisions and am satisfied that my concerns have been fully addressed. In my opinion, the paper is now suitable for acceptance in the current form.

Reviewer #4

(Remarks to the Author)

From my original review only one point remains open.

The authors have now shown increased epithelial thickness and goblet cell number in the Af model, however, no data demonstrating improvement in these readouts after DC cell manipulation was included, which leaves open the question whether Mgl2 cDC2s mediates allergic airway inflammation and translates into protection of pathological changes.

Minor comments:

From the recently inserted histology data and description, it is unclear which size airways were taken for epithelial thickness measurements and Masson's trichrome quantification? For the later it currently reads that the entire lung was quantified for collagen levels, and not specifically the airways. In all cases the analysis should be restricted to the airways of 100 μm – 300 μm . Please correct.

I would suggest to include the Dot plot representing the results of gene-set enrichment analysis (Supplementary Figure 12A) into Figure 4, which would give the reader more immediate insights into the cellular markers and associated pathways for the different DC/Mo populations.
